



# Characterization of the Particle Emission from Ships Operating at
# Sea Using Unmanned Aerial Vehicles
Tommaso F. Villa[1], Reece Brown[1], E. Rohan Jayaratne[1], L. Felipe Gonzalez[2], Lidia Morawska[1], Zoran
D. Ristovski[1*]
[1] International Laboratory for Air Quality and Health (ILAQH), Queensland University of Technology (QUT), 2 George St,
Brisbane QLD 4000
[2]Australian Research Centre for Aerospace Automation (ARCAA), Queensland University of Technology (QUT), 2 George
St, Brisbane QLD 4000
*Correspondence to*: Zoran D. Ristovski (z.ristovski@qut.edu.au)
**Abstract.** This research demonstrates the use of an unmanned aerial vehicle (UAV) to characterize the gaseous ($CO_2$) and
particle (10 - 500 nm) emissions of a ship at sea. The field study was part of the research voyage "The Great Barrier Reef as
a significant source of climatically relevant aerosol particles" on-board the RV Investigator around the Australian Great
Barrier Reef. Measurements of the RV Investigator exhaust plume were carried out while the ship was operating at sea, at a
steady engine load of 30%.
The UAV system was flown autonomously using several different programmed paths. These incorporated different altitudes
and distances behind the ship in order to investigate the optimal position to capture the ship plume. Five flights were
performed, providing a total of 27 horizontal transects perpendicular to the ship exhaust plume. Results show that the most
appropriate altitude and distance to effectively capture the plume was 25 m above sea level and 20 m downwind.
Particle number (PN) emission factors (EF) were calculated in terms of number of particles emitted (#) per weight of fuel
consumed (Kg fuel). Fuel consumption was calculated using the simultaneous measurements of plume $CO_2$ concentration.
Calculated $EF_{PN}$ were between 9.19 x $10^{14}$ and 5.15 x $10^{15}$ #·(Kg fuel)$^{-1}$ These values are in line with those reported in the
literature for ship emissions ranging from 0.2 6.2 x $10^{16}$ #·(Kg fuel)$^{-1}$ to 6.2 x $10^{16}$ #·(Kg fuel)$^{-1}$.
This UAV system successfully assessed ship emissions to derive emission factors (EFs) under real world conditions. This is
significant as, for the first time, it provides a reliable, inexpensive and accessible way to assess and potentially regulate ship
emissions.
**1.      Introduction**
Shipping is the most significant contributor to international freight, with almost 80% of the worldwide merchandise trade by
volume transported by ships in 2015 (UNCTAD 2015). Emissions from this transportation mode are a significant contributor
to air pollution, both locally and globally. Ships are a major pollutant source in areas surrounding harbours (Viana et al.
2014), with over 70% of emissions reaching 400 km inland (Fuglestvedt et al. 2009). In 2012 exhaust from diesel engines,
the predominant source of ship power, was classified as a group 1 carcinogen by the International Agency for Research on
Cancer (IARC). In 2007, pollution from ship exhaust was found to be responsible for approximately 60,000
cardiopulmonary and lung cancer deaths worldwide annually (Corbett et al. 2007a). Such emissions are also a strong climate
forcing agent, contributing to global warming through the absorbance of solar and terrestrial radiation (Hallquist et al. 2013a;
Lack et al. 2011; Winnes et al. 2016).
Despite these findings, emissions from shipping have consistently been subject to less regulation than those of land-based
transport with ship emissions in international waters remaining one of the least regulated parts of the global transportation
system (Cooper 2001; 2005; Corbett and Farrell 2002; Corbett and Koehler 2003; Eyring et al. 2005; Streets et al. 1997;
USEPA-OTAC 2012). Currently, no specific restrictions for ship-emitted particulate matter (PM) exist, with the only
regulated pollutants being NOx and $SO_2$. The International Maritime Organization (IMO) recently revised the regulation of



these gaseous pollutants through the Annex VI of the International Convention for the Prevention of Pollution from Ships –
the Marine Pollution Convention (MARPOL). The IMO expected that these regulations would lead to an indirect decrease in
particle number (PN) concentration due to the reduction of NOx emissions and the use of fuel with lower sulphur content
[14]. However, it has been found that the use of some low sulphur fuels lead to increased PN concentrations at lower engine
loads (Anderson et al., 2015), which stresses the importance for regulation specifically addressing particulate matter (PM).
The majority of emitted PM is in the ultrafine size range, < 0.1 μm, which have been demonstrated to have a particularly
significant impact on health and the environment (WHO 2013 ). However, due to the lack in regulation, ultrafine particles, in
terms of PN concentration, emitted from ships have remained unassessed in real world conditions. Quantifying PN
concentration is critical to improve our understanding of shipping's impact on health and climate (Anderson et al. 2015;
Blasco et al. 2014; Chen et al. 2005; Cooper 2001; Corbett and Farrell 2002; Corbett et al. 2007b; Isakson et al. 2001;
Mueller et al. 2015; Reda et al. 2015; Ristovski et al. 2012; Williams et al. 2009). To achieve this, wide-scale evaluation of
ship emission factors (EFs) is necessary. EFs are commonly expressed as the amount of pollutant (x) emitted per unit mass
of fuel consumed $g(x). (Kg fuel)^{-1}$. Different methods have been used to investigate ship EFs, including laboratory test-bench
studies, on-board measurements, and measurement of ship emission plumes.
Test-bench studies (Anderson et al. 2015; Kasper et al. 2007; Mueller et al. 2015; Petzold et al. 2008; Petzold et al. 2010;
Reda et al. 2015) have been used to characterize emissions from different engines at various loads in laboratory conditions.
However, engine performance and emissions have been shown to be different in real world operations when compared to
laboratory studies. This calls for measurements of ship emissions in-situ to collect reliable data for EF calculations (Agrawal
et al. 2008; Blasco et al. 2014; Murphy et al. 2009). To date, only a few studies have been undertaken on-board ships to
calculate real emission factors (Hallquist et al. 2013b; Juwono et al. 2013). This is attributed to the prohibitive costs and time
commitments of setting up and maintaining on-board measurement equipment on commercial ships. Airborne ship plume
measurements (Balzani Lööv et al. 2014; Beecken et al. 2014a; Berg et al. 2012; Cappa et al. 2014; Lack et al. 2008; Lack et
al. 2009; Pirjola et al. 2014a; Schreier et al. 2015; Sinha et al. 2003; Westerlund et al. 2015) offer an alternative method of
in-situ measurements without requiring on-board monitoring stations. In the past the cost, the significant difficulties in
deployment of these systems, and the risk for manned aircrafts have limited their feasibility. However, this has recently
changed with the rapid advances being made in commercially available Unmanned Aerial Vehicle (UAV) technology.
Hexacopter UAVs have seen a wide scale increase in industry and research applications due to their ease of use and
comparatively low cost (Brady et al. 2016; Gonzalez et al. 2011; Malaver Rojas et al. 2015). Used in conjunction with air
monitoring equipment, these systems provide, for the first time, the ability to perform relatively simplistic and cost-effective
airborne measurements of ship emissions. However, to date no studies have reported the use of a UAV system capable of
collecting data to calculate the EF of PN concentration for ships at sea.
This research utilized a customized hexacopter UAV carrying instruments for PN concentration and $CO_2$ measurements to
derive $EF_{PN}$. The UAV system was deployed from the RV Investigator research vessel while at sea. Autonomous
measurements of the RV investigators exhaust plume were taken over several flights at various altitudes and distances from
the ship. Data collected was used to optimize the sampling flight path and successfully quantify the RV investigators EF for
PN concentration.
**2.       Methodology and Measurement system**
Measurements were conducted as part of the research voyage "The Great Barrier Reef as a significant source of climatically
relevant aerosol particles" aboard the RV Investigator research vessel over a two day period of the 13 and 14 October 2016
(day 1 and day 2). Measurements of PN and $CO_2$ concentration emitted by the RV Investigator were taken using a PN and
$CO_2$ monitor mounted on a customized DJI EVO S800 hexacopter UAV (DJI 2014).





## 2.1. The RV Investigator and the voyage

The RV Investigator is a sophisticated ocean research vessel configured to enable a wide range of world class atmospheric, biological, goescience and oceanographic research. The vessel is 94 m long, has a gross weight of 6,082 tons, a fuel capacity of 700 tons of ultra-low sulphur diesel fuel. It is powered by three 9 cylinder 3000 kW MaK diesel engines, each coupled to a 690V AC Generator. Ship propulsion is achieved using two 2600 kW L3 AC reversible propulsion motors powered by these generators. The RV Investigator can host up to 30 crew members and 35 researchers for a maximum voyage period of 60 days with at a maximum cruising speed of 12 knots.

A suite of instrumentation for atmospheric research is available on the RV Investigator. This includes a radar system capable of collecting weather information within a 150 km radius of the vessel, and instruments measuring: sunlight parameters; aerosol composition, particle concentration and size distributions; cloud condensation nuclei; gas concentrations; and various other components of the atmosphere. These instruments are housed inside two dedicated on-board laboratories for aerosol and for atmospheric chemistry research. An atmospheric aerosol sample is continuously drawn into the laboratories for analysis through a specialized inlet fitted to the foremast of the ship. Of particular interest to this study, the ship contains a PICARRO (PICARRO Inc., Santa Clara, California, USA) G2401 analyser (Inc. 2017) that continuously measures $CO_2$, CO, $H_2O$ and $CH_4$. It has an operation range between 0-1000 ppm and a parts-per-billion sensitivity (ppb) for $CO_2$.

The two day UAV measurement study was possible as part of the RV Investigator voyage "The Great Barrier Reef as a significant source of climatically relevant aerosol particles", which started in Brisbane on the 28th of September 2016. The ship was used as both: a floating platform to allow launch and recovery of the UAV system; and as the source of an exhaust plume measured by the UAV system for EF calculation. During a several day stationary period on the Great Barrier Reef off the coast of Australia, it was possible to measure the ship plume under stable real world conditions over two consecutive days. One of the three ship engines was maintained at a steady engine load of 25 – 30 % of the maximum engine power during all measurements.

## 2.2. UAV system

Measurements of PN and $CO_2$ concentrations in the ship plume were performed using two commercial sensors mounted on-board a hexacopter UAV. The UAV used (Figure 1) is a composite material S800 EVO manufactured by DJI (DJI 2014). The UAV is 800 mm wide and 320 mm in height, with an unloaded weight of 3.7 kg. Minimum and maximum take-off weights are 6.7 kg and 8 kg, respectively. The UAV contains a 16000 mAh LiPo 6 cell battery, which provides a hover time of approximately 20 min when operating at minimum take-off weight. The telemetry range of the UAV is 2 km, which was adequate to cover the desired sampling area (See Figure 2).

The payload consisted of a PN concentration and a $CO_2$ monitor mounted on-board underneath the UAV. Careful placement of the payload was required to prevent flight issues caused by an altered centre of gravity. Also included was a carbon fiber rod, which extended outward horizontally from the UAV. The sampling lines for the monitors were attached to the end of this rod to ensure that measurements were not affected by the downwash of the UAV rotors. The total weight of the payload was (1.2 kg), which allowed the UAV system to fly for 12-15 min before landing at the home point (A) (See Figure 2).

The S800 was used in conjunction with the DJI Wookong autopilot. The software provides an intuitive and easy to use interface where autonomous flight paths can be planned, saved, and uploaded into the UAV. In addition to this, the ground station allows for continuous, real-time monitoring of the status of the UAV during operation; which includes its longitude, latitude, altitude, waypoint tolerance and airspeed.

The DJI S800 was chosen for this study because it is designed to operate under the 20 kg all up weight (AUW) class of UAV. This reduces operational costs and avoid subjection to the tighter regulations of larger platforms. Small UAV cannot be operated above any person, or closer than 30 m of populated areas, houses and people. Furthermore, current Civil Aviation Safety Australia (CASA) regulations restrict the use of small UAV (2 and 20 kg) to visual line-of-sight daylight





operation, with a maximum altitude of approximately 120 m and within a radius of 3 nmi of an airport. UAVs in this
category are not permitted for research unless the research institution has been granted a permit exception. These exceptions
can be granted if the institution in question has or collaborates with an UAV operation team who must have: an experienced
UAV pilot who is also radio controller specialist; a license for commercial UAV operation; and appropriate liability
insurance . Queensland University of Technology (QUT) has an unmanned operator certificate and four pilots who have
UAV controller licenses.
**2.2.1.    Instrumentation**
**2.2.1.1.    Instrumentation for PN concentration**
This study measured PN concentration using a Mini Diffusion Size Classifier (DISCmini), developed by the University of
Applied Sciences, Windisch, Switzerland (Fierz et al. 2008). The DISCmini is a portable monitor used to measure
concentration of particles in the 10-500 nm diameter size range, with a time resolution of up to 1s (1 Hz). It can measure PN
concentrations between $10^3$ and $10^6$ N/cm$^3$. Measurement accuracy is dependent upon the particle shape, size distribution,
and number concentration. The advantages of using the DISCmini are its relatively small dimensions (180 x 90 x 40 mm),
low weight (640 g, 780 g with the sampling probe, Figure 1) and long battery life of up to 8 hrs. These
characteristics allow it to be easily integrated on the UAV.
**2.2.1.2. Instrumentation for CO2 concentration measurements**
A TSI (TSI, Shoreview, Minnesota, United States) IAQ-calc 7545 model was chosen to measure $CO_2$ concentrations. Its
sensor is based on a dual-wavelength NDIR (non-dispersive infrared) with a sensitivity range between 0 to 5,000 ppm and an
accuracy of ±3.0% of reading or ± 50 ppm (whichever is greater). The measurement resolution is 1 ppm with a maximum
time resolution of 1s. Similar to the DISCmini, the advantages of using the IAQ-calc are: its small dimensions (178 x 84 x 44
mm); low weight (270 g, with batteries, significantly lower than the DISCmini), and a battery life of 10 hours.
The readings of the IAQ-clac for $CO_2$ were compared with those measured by the on-board PICARRO G2401 analyser.
Both the DISCmini and the IAQ-calc were tested and calibrated in the laboratory prior to the commencement of the
measurements (Figure S1 in the Supplementary Material). All data was logged with a 1 s time interval.

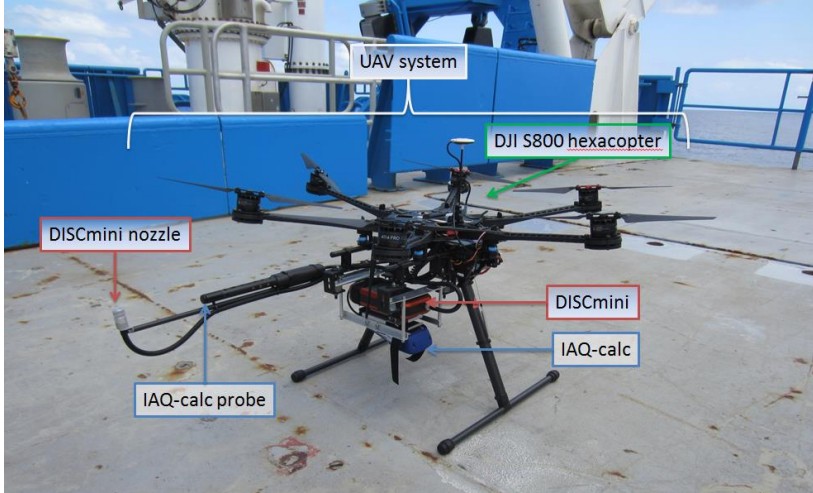


**Figure 1. The UAV system with the on-board instrumentation: the DISCmini and the IAQ-calc.**



### 2.3.    Meteorological data


Meteorological data (including air temperature, relative humidity, atmospheric pressure, wind speed and direction) were
recorded by the RV Investigators on-board instrumentation during the entire voyage with a 60 s time interval, 24/h a day.

### 2.4.    Study design


During the two measurement days of this study, the vessel was heading into the wind whilst idling the UAV missions at sea.
This positioning caused the exhaust plume to extend downwind, directly behind the ship. The UAV system was launched off
the back deck, autonomously sampling at varying altitudes and distances into the downwind plume. Flight speed of the UAV
was 1.5 m/s, the minimum for the S800.
Day 1 was used to optimise the study design, focusing on finding the flight path most suitable to capture the ship plume.
Figure 2 shows the programmed flight path, which consisted of a continuous flight beginning at a distance (D) and from an
altitude (H) above the surface. Point A, located on the back deck of the RV Investigator, represents the 'home point'. In
UAV terminology this refers to the position where the UAV system takes off and lands. The UAV system was programmed
to move  horizontally by a distance (2d), perpendicular to the ship, then climb vertically for 10 m (h) before flying in the
opposite horizontal direction for the same distance (2d). The UAV was then programmed to climb another 10 m (h) before
repeating this pattern until the UAV reached an altitude of 65 m above the ocean. During day 1, the UAV system followed
three different flight paths, each one with both a different distance D behind the ship (20, 50 and 100 m), and a different
horizontal distance 2d (50, 100 and 150 m).
The optimised flight path for day 2 started 20 m behind the ship and 25 m above the surface, with no altitude variation. The
UAV path was limited to a continuous horizontal flight of 50 m (2d) at steady speed of 2 m s$^{-1}$. This path and flying speed
allowed up to 4 horizontal transects to capture the ship plume.

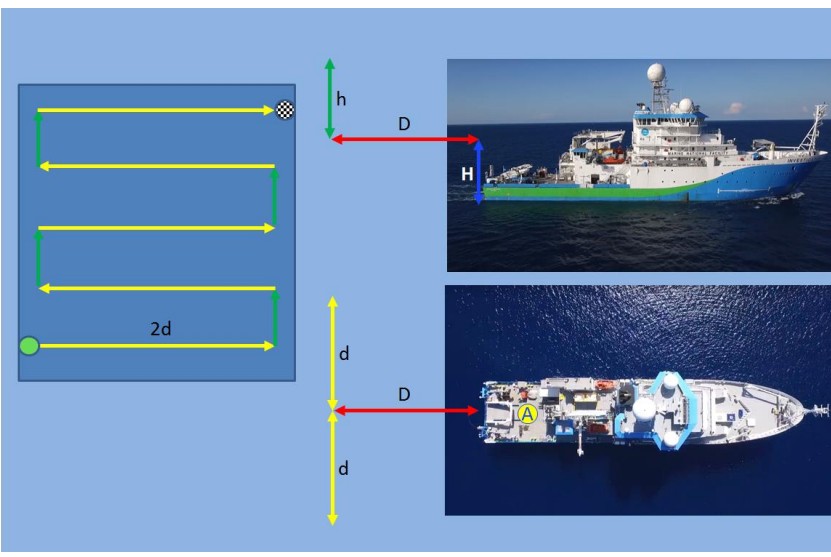

**Figure 2. Flight path used to capture the plume: H - height from the ocean, D – distance behind the ship to the flight beginning point, h – rising altitude after the horizontal transect, 2d – full length of the horizontal transect**

### 2.5.    Experimental procedure

The UAV can fly either manually or autonomously. As a safety precaution, every take-off and landing was performed using
the manual flight mode. Once in the air, the UAV was switched to autonomous flight mode, allowing the platform to follow
the pre-programmed flight path discussed in the previous section. The flight path consisted of waypoints, which are three-





dimensional GPS points that dictate the position of the UAV along the fight path. The waypoints and flight plans for each
flight were programmed using the aforementioned DJI Wookong ground station software. The DISCmini and the IAQ-calc
were fitted on the underside of the UAV at the beginning of each measuring day. Five flights were performed across the two
measurement days, providing a total of 27 horizontal transects perpendicular to the ship's exhaust plume.
**2.6.    Emission factors**
The calculation of an emission factor for particle number concentration ($EF_{PN}$) from the collected ship plume measurements
was performed using Eq. (1). This method has previously been used for ship (Westerlund et al. 2015), road vehicle (Hak et
al. 2009) and aircraft (Mazaheri et al. 2009) emissions. The measured values of PN concentration were related to the amount
of fuel consumed by the engine in question through the use of the simultaneous measurements of $CO_2$ concentration taken by
the UAV. This was achieved by using a published value for a ship emission factor of $CO_2$ ($EF_{gas}$) of 3.2 Kg $CO_2$ (Kg fuel)$^{-1}$
(Hallquist et al. 2013b; Hobbs et al. 2000) .
Eq.(1).
$$EF_{PN} = \frac{\Delta PN}{\Delta gas} \text{ x } EF_{gas}$$
(1)

The $\Delta PN$ and $\Delta gas$ in Eq. (1) represent changes in the measured particle number and $CO_2$ concentrations, respectively.
Background concentrations of PN and $CO_2$ were subtracted and $EF_{PN}$ was calculated by integrating the peak plume
concentration measured by the DISCmini and IAQ-clac mounted on the UAV system; which is defined as the average
concentration measured by the DISCmini and IAQ-calc outside the ship plume.

**3.    Results and Discussion**
**3.1.    Meteorological and Investigator data**
Wind conditions were very stable during both day 1 and day 2, following one main pattern for the entire flight time. The
wind speed ranged from 3 - 13 m s-1. The wind direction was predominantly from the NE during day 1 and ESE during day

200 2.

The wind rose graphs in Figure 3a and 3b illustrate the wind data recorded with the on-board weather instrumentation during
all horizontal transects flown during day 1 and 2 respectively. The prevalent wind direction was ESE, which corresponded to
the heading of the RV Investigator (indicated by the rose triangle).
The wind direction changed occasionally to E during the flight, causing the UAV to fail to capture the RV Investigator
plume during some transects. As a result, 2 of the 8 horizontal transects collected on day 2 were excluded from the analysis.



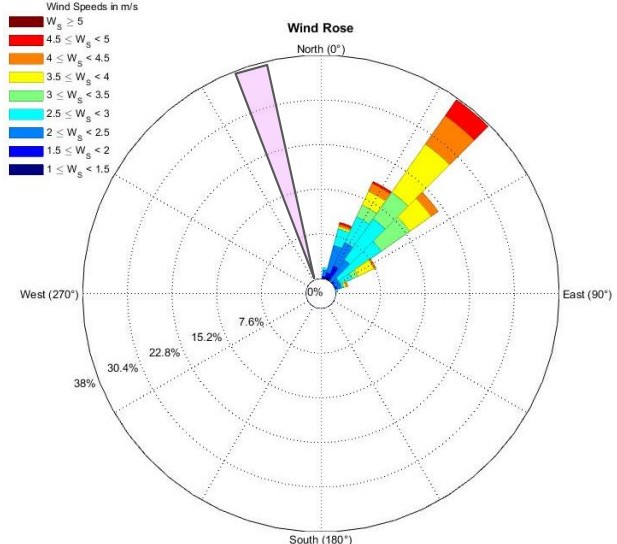


**Figure 3a – Wind rose showing wind speed and direction during day 1. Rose triangle shows RV Investigator direction during the**

**measurements.**


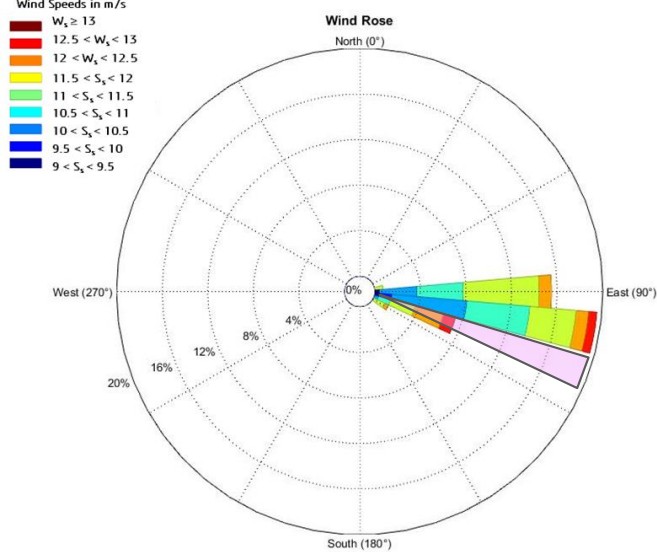


**Figure 3b – Wind rose showing wind speed and direction during day 2 optimized flight. Rose triangle shows RV Investigator**

**direction during the measurements.**

**3.2.    UAV system horizontal transects inside and outside the plume**
The UAV system acquired data for a total of 27 horizontal transects for day 1 and day 2. Data was collected at altitudes
between 25 m and 65 m above the water surface. During day 1 the plume was captured once when the UAV was at 25 m





altitude and 20 m downwind of the ship; and again at both 25 and 35 m altitude 100 m downwind of the ship. These
observations lead to the optimized flight used on day 2, which started downwind at 25 m above the surface and 20 m behind
the ship. On day 2 the UAV system successfully captured the plume during 6 of the 8 transects performed. Across the two
days this lead to a total of 9 transects that captured the plume and which have been considered for discussion, shown in
Table 1.

| Measuring day | Altitude | Distance behind the Investigator | Number of transects |
|---|---|---|---|
| Day 1 | 25 m | 20 m | 1 |
| *Day 1 | 25 m | 100 m | 1 |
| Day 1 | 35 m | 100 m | 1 |
| Day 2 | 25 m | 20 m | 6 |


**Table 1 – Specifications of the transects considered for the data analysis. The (*) indicates the transect of Day 1 of which PN**
**concentration and $CO_2$ profiles are presented in Figure 4.**

Figure 4 shows the PN concentration and $CO_2$ profiles, collected during two (a; b) transects on day 2, and (c) during one
transect of day 1 (Spec. in Table 1, Day1*).
The PN concentration profiles for the (a) and (b) transects in Figure 4 show that the concentration varied by five orders of
magnitude between the outside and inside the plume, while the $CO_2$ profiles show an increase up to 140 ppm above the
background.
The profiles in (c) show that the PN concentration was four orders of magnitude greater inside the plume at 100 m behind the
ship and that the $CO_2$ concentration was up to 100 ppm higher inside the plume.





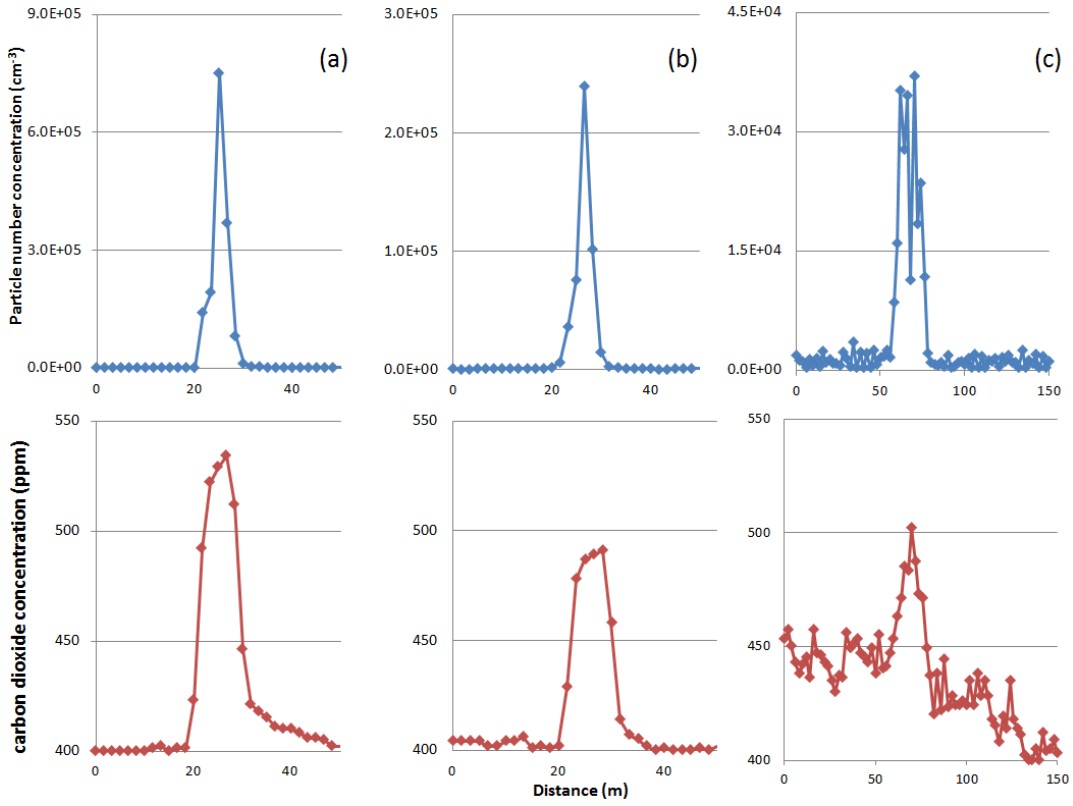


**Figure 4 – (a) and (b) show the PN concentration and CO₂ profiles collected at 20 m behind the ship 25 m above the surface during**


**one of the flight in day 2. (c) shows the PN concentration and CO₂ profiles collected during flight 3 of day 1 at 100 m behind the**


**ship, 25 m above the surface.**



Figure 4 (a) and (b) both show transects at 25 m altitude and 20 m behind the ship. Both the PN concentration and $CO_2$
measurements show clear, single peaks as the UAV crosses the plume. However, the maximum PN concentrations measured
in (a) ($7.5 \times 10^5$ #.cm$^{-3}$) are approximately three times greater than those in (b) ($2.4 \times 10^5$ #.cm$^{-3}$). Furthermore, the $CO_2$
measurements between (a) and (b) have a difference of (43ppm). As the ship engine remained under steady load throughout
these measurements, the variations between (a) and (b) can be attributed to several factors which reduce the effectiveness of
the UAV transect for capturing the plume. Slight changes in ambient conditions such as temperature, wind direction and
intensity will alter the path of the plume as it moves away from the ship. The UAVs automated flight path cannot account for
these variations. Therefore, the degree to which the UAV enters the plume, and thus the concentrations it measures, will be
different on each transect. Both $CO_2$ and PN concentration measurements will be similarly affected by this variance.
However; it is expected that this will contribute to the calculated error margin of the final result.
In comparison to Figure 4 (a) and (b), the graphs in (c) show substantially less defined, wider peaks with lower pollutant
concentrations. This is attributed to a difference in flight paths, with Figure (c) representing data from a transect 100 m
behind the ship; whilst (a) and (b) were performed 20 m behind the ship. As the plume travels away from the ship it will
begin to turbulently mix with the surrounding air mass; causing concentrations to decrease and the plume to broaden as the
pollutants spread into the atmosphere.





A potential benefit of the 100 m transect is that it provides more data points inside the plume when compared to the 20 m
transect. However, there are clear variations in the measurements across the plume, indicating that the plume was not
homogenous at this distance. This could be due to localized perturbations in the wind causing inconsistent mixing with the
surrounding air mass. Furthermore, the $CO_2$ measurements do not follow the PN concentration measurements; with the peak
being significantly broader and not returning to its expected background value of around 400 ppm. These issues indicate that
more distant measurements, whilst providing more data points, potentially provide less accurate data for the calculation of
emission factors. More accurate transect measurements could be achieved by slowing the UAV flight speed for transects
closer to the emission source. However, this was not possible in this study as the S800 hexacopter UAV was flown at its
minimum speed of 1.5 m/s during all transects.

### 263   3.3.      PN Emission Factors

$EF_{PN}$ values were calculated relative to the fuel consumption using the fuel combustion derived plume $CO_2$, (Eq. 1) and the
data from the nine transects listed in Table 1.
$\Delta PN$ was calculated by integrating the peak plume concentration (average of five data points) measured by the DISCmini,
after subtraction of the background concentration. Background concentration is defined as the concentration (average of five
data points) measured outside the plume. The same calculation was made to obtain the $\Delta CO_2$.
Table 2 shows, for each of the 9 transects, where the plume was captured, the measured concentration values of $\Delta PN$
and $\Delta CO_2$, in $Kg$ per cubic meter, and the calculated $EF_{PN}$.

| Day | Plume captured (distance and altitude) | $\Delta PN \ m^3$ | $\Delta CO_2$ (Kg) | $EF_{PN}$ |
|---|---|---|---|---|
| 1 | 20 m; 25 m | $1.94 \times 10^{11}$ | $1.21 \times 10^{-4}$ | $5.11 \times 10^{15}$ |
|   | 100 m; 25 m | $2.83 \times 10^{10}$ | $9.86 \times 10^{-5}$ | $9.19 \times 10^{14}$ |
|   | 100 m; 35 m | $4.72 \times 10^{10}$ | $8.88 \times 10^{-5}$ | $1.70 \times 10^{15}$ |
| 2 | 20 m; 25 m | $3.07 \times 10^{11}$ | $2.30 \times 10^{-4}$ | $4.27 \times 10^{15}$ |
|   | 20 m; 25 m | $9.18 \times 10^{10}$ | $1.41 \times 10^{-4}$ | $2.08 \times 10^{15}$ |
|   | 20 m; 25 m | $4.81 \times 10^{10}$ | $9.55 \times 10^{-5}$ | $1.61 \times 10^{15}$ |
|   | 20 m; 25 m | $1.78 \times 10^{11}$ | $1.94 \times 10^{-4}$ | $2.94 \times 10^{15}$ |
|   | 20 m; 25 m | $8.05 \times 10^{10}$ | $8.29 \times 10^{-5}$ | $3.11 \times 10^{15}$ |
|   | 20 m; 25 m | $7.46 \times 10^{10}$ | $1.21 \times 10^{-4}$ | $1.98 \times 10^{15}$ |


**Table 2 – $\Delta PN$ and $\Delta CO_2$ concentration emission/rate of the RV Investigator and ccalculated Emission Factors for PN.**

The $\Delta PN$ and $\Delta CO_2$ values were plotted and correlated against each other as shown in Figure 5. $\Delta PN$ and $\Delta CO_2$ were found
to have a good linear relationship with an $R^2$ value of 0.7529.





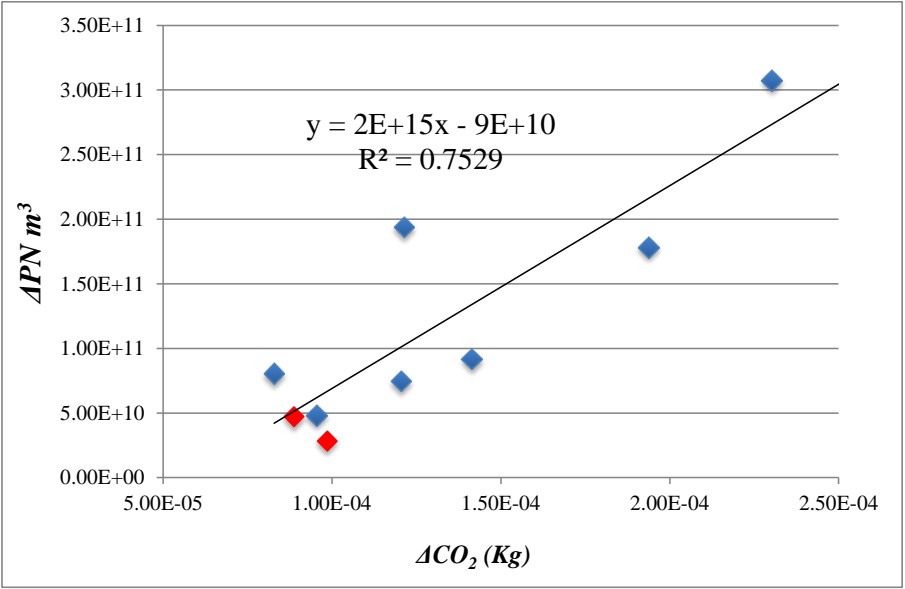


**Figure 5 –$\Delta PN$ and $\Delta CO_2$ for the nine transects considered for the data analysis. Red markers indicate the measurements taken at**
**100 m behind the ship.**

The calculated $EF_{PN}$ values for the RV Investigator ranged from 9.19 x $10^{14}$ to 5.11 x $10^{15}$ #·(Kg fuel)$^{-1}$. The two 100 meter
transects provided the lowest two emission factors measured (9.19 x $10^{14}$ #·(Kg fuel)$^{-1}$ and 1.70 x $10^{15}$ #·(Kg fuel)$^{-1}$). This is
likely a consequence of the noted differences between the plume measurements of the 20 and 100 m transects. The clear
distinction between the background and the plume measurements of the 20 m transect indicate that the $EF_{PN}$ calculated using
them will be more representative of the RV Investigator emissions at 30% engine load. Therefore, the 100 m transects were
discounted from the calculation of the mean $EF_{PN}$ and the corresponding standard error. These values were calculated as 3.0
x $10^{15}$ ± 0.5 x $10^{15}$ #·(Kg fuel)$^{-1}$. As presented in Table 3, this value is comparable with those reported in the literature for
cruise and cargo ship plumes; which range from 0.2 x $10^{16}$ to 6.2 x $10^{16}$ #·(Kg fuel)$^{-1}$ (Alföldy et al. 2013; Beecken et al.
2014b; Jonsson et al. 2011; Juwono et al. 2013; Lack et al. 2011; Pirjola et al. 2014b; Sinha et al. 2003; Westerlund et al.

289      2015).

The calculated $EF_{PN}$ for the Investigator were lower compared to those reported by Beecken at al. (Beecken et al. 2014a) for
passenger ships while accelerating (0.91 ± 0.18 x $10^{16}$ # (Kg fuel)$^{-1}$). However, the RV Investigator measurements were
undertaken whilst its engine was under 30% load. Accelerating ships will typically be under higher engine loads and hence
have a correspondingly higher $EF_{PN}$ (Westerlund et al. 2015), which explains part of this discrepancy. Furthermore, the RV
Investigator is a sophisticated modern vessel built for use in regions such as Antarctica. As such, it is design to have high
efficiency engines, a diesel-electric energy generation system, and uses refined, ultra-low sulphur diesel fuel. These factors
lead to the RV Investigator being more efficient and less polluting than most other ships at sea. This explains why the results
of this study are comparable to the lower end of those found in the literature.
The RV investigator also uses low sulphur content diesel fuel which is similar in quality to the fuels used in the ground
transport industry. In fact, the results presented here were comparable to those for in-land transportation, ranging from 4.8 x
$10^{14}$ (25% engine load) to 7.2 (100% engine load) x $10^{15}$ # (Kg fuel)$^{-1}$ (Jayaratne et al. 2009). The calculated values for the




RV Investigators $EF_{PN}$ are also close to data for commercial aircrafts during landing and taxing, which range from 4.16 to
$7.74 \pm 1.46 \times 10^{15}$ # (Kg fuel)$^{-1}$ (Mazaheri et al. 2009).

| Reference | Measuring Platform | EF (PN) | Number of ships | Location |
|---|---|---|---|---|
| This Study | Unmanned Aerial Vehicle | $0.3 \times 10^{16}$ | 1 | Open water |
| Westerlund et al. (2015) | Land based | $2.35 \pm 0.20 \times 10^{16}$ | 154 | Harbor, Ship Channel |
| Beecken et al. (2014) | Airborne | $1.8 \pm 1.3 \times 10^{16}$ | 174 | Open water |
| Pirjola et al. (2014) | Land based | $0.32 \times 10^{16}$ | 11 | Harbor, Ship Channel |
| Alföldy et al. (2013) | Land based | $0.8 \times 10^{16}$ | 497 | Harbor |
| Juwono et al. (2012 | On board | $0.22 \times 10^{16}$ | 2 | Harbor, Ship Channel |
| Jonsson et al. (2011) | Land based | $2.55 \pm 0.11 \times 10^{16}$ | 734 | Harbor |
| Lack et al. (2011) | Airborne | $1.0 \pm 0.2 \times 10^{16}$ | 1 | Open water |
| Sinha et al. (2003) | Airborne | $6.2 \pm 0.6 \times 10^{16}$ | 2 | Open water |


**Table 3 – Comparison of the Emission Factor for the RV Investigator found in this study with other relevant values found in**
**literature.**
**4.        Summary and conclusion**
The UAV system used in this study successfully measured PN and $CO_2$ concentrations from the exhaust plume of the RV
Investigator whilst operating at sea. Several different flight paths were tested and an optimal transect flying perpendicular to
the plume at a distance of 20 meters from the ship was adopted. The $EF_{PN}$ calculated for the RV investigator ranged from
$9.19 \times 10^{14}$ to $5.11 \times 10^{15}$ #·(Kg fuel)$^{-1}$ relative to both consumed fuel and engine load. This $EF_{PN}$ was within the lower end
of values reported in literature, thus validating the novel UAV system for this application.
In comparison with other methods, the UAV system presented provides a cost effective and accessible solution for the rapid
measurement and quantification of ship emissions. Its ability for deployment both in harbour and at sea, coupled with the
possibility of altering its flight path to account for variances in wind conditions; gives this UAV system a distinct advantage
over ground based and manned aerial vehicles. Furthermore, the UAV can sample considerably closer to the plume emission
source than other methodologies, providing more accurate measurements for the calculation of $EF_{PN}$.
These attributes indicate that this UAV system provides a basis for wide-scale quantification of ultrafine particle emission
factors from commercial shipping. This is critical to improve our understanding of shipping's impact on climate and health.
Furthermore, it will both inform regulatory bodies, and provide them with the tools to monitor emissions in harbours and at
sea.
**4.1.    Recommendations**
The possibilities of this UAV system extend far beyond what is described here. This study is intended as both: a proof of
concept; and to provide useful information both for the future of this project, as well as any other UAV sampling systems





being developed. The instruments on-board this system were used for the measurement of PN and $CO_2$ concentrations in
order to calculate $EF_{PN}$. However, this methodology could also be expanded to measure other important ship emission
factors, including $NO_X$ and volatile organic compounds (VOCs).
Further possibilities and potential improvements can also be made to the plume transect sampling method used here. The
sampling error could be reduced by collecting more data points inside of the plume. One method to achieve this would be to
find an optimal transect distance which provides the broadest plume cross-section, whilst also providing a clear
differentiation between plume and the surrounding air mass. An alternative approach would be the use of a different UAV
with a lower minimum operational speed to increase the time of the plume transect. Other study possibilities include:
comparisons between $EF_{PN}$ for different loads both in the harbour and at sea, and investigations into the use of a single flight
to transect multiple ship plumes.
The transect-based sampling approach provides researchers with a relatively simple method of capturing data inside the
plume. The principal flaws with this method are that there is no guarantee that the plume will be captured during a transect,
and the degree to which the UAV enters the plume can vary between transects. A potential answer to these issues is a non-
transect based approach in which the UAV system is made to hover inside the plume for a given period of time, ensuring
data is collected. This also allows for the collection of many more data points inside the plume, ensuring accurate and
repeatable data. Despite these advantages this method has proven to be challenging as it is difficult to verify whether the
UAV is within the plume, when it is not visible to the naked eye especially in variable wind conditions. A potential solution
is the implementation of sensors and instrumentation which transmit data to the ground station in real time. Using this data
as a feedback mechanism, it would be possible to orient the UAV position so it hovers within the plume, ensuring that more
accurate and repeatable data is collected on every flight.
**Acknowledgements**
The authors would like to acknowledge the ARCAA Operations Team (Dirk Lessner, Gavin Broadbent) who operated the
Unmanned Aerial Vehicle (S800). This research was supported by the Australian Research Council Discovery Grant
DP150101649 and the Marine National Facility. The authors would like to thank the Captain and the crew of the RV
Investigator as well as the on board MNF support staff as without their support and effort this research would not have been
possible.

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
