# Peer review of "Characterization of the Particle Emission from Ships Operating at 2 Sea Using Unmanned Aerial Vehicles"

_Atmospheric Measurement Techniques, 2017_

## Referee Comment (RC1) · Anonymous Referee #1 · 27 Oct 2017

Line 22: There is a typo with the first emission factor given.

L24: The authors indicate that they have demonstrated a "reliable, inexpensive and accessible" way of measuring ship emissions. The measurements here required the UAV being deployed from on board the ship. This seems potentially quite limiting. It would be useful if the authors were to rethink the concept of "accessible." I do not dispute that the method here has potential. But, it has not been demonstrated that it is an "accessible" method, especially given the need to optimize the flight path before performing measurements.

L83: I suggest that both "sophisticated" and "world class" be removed. There is no

need for these superlatives, nor are they justified by the description.

L120-129: While an interesting discussion, at the end it was a little unclear what was specifically done here.

Substantailly more information regarding calibration and testing of the DISCmini and the IAQ-calc are needed. The supplemental has no information on the CO2 comparison. This should be added. For the particle comparison, the authors should indicate the measurement conditions. As they note, the calibration depends on the assumed particle size distribution. What was chosen for calibration? Was this just ambient air? Particles produced from an atomizer? Are the calibration particles relevant to the particles in the plume in terms of the size distribution? Were the DISCmini concentrations corrected to account for the difference in slopes in Fig. S1? Is that what is meant by calibration, or are the instruments just being compared? How is uncertainty estimated? A lot more information is needed for both CO2 and particles.

Eqn. 1: The authors use the integrated peak concentrations to calculate the ratio between delta values and the EF. From Fig. 4 it is evident that the CO2 plume is broader than the PN plume. The authors should consider discussing this issue in the context of how it impacts their EF estimates.

L240: Here, the authors focus on differences in absolute values. Such differences can result for a variety of reasons. What really matters, though, is how different the derived EF values are. I suggest that the authors bring the EFs for these plumes into the discussion.

Table 2 and Fig. 5: The units given for CO2 are not correct. This must be kg/mˆ3. It is not possible to simply have kg as the units, since the volume is not known. Also, if the units are not kg/mˆ3, then the units on the derived EFs will not make sense.

Table 2 vs. Fig. 5: There seems to be an inconsistency. The slope from Fig. 5 can be converted to an EF by multiplying by 3.2 kg CO2/kg-fuel. This yields 6.4e15

[Figure]

particles/kg-fuel. But, the average from Table 2 is only 2.6e15 particles/kg-fuel. These should be closer. The difference may be because the authors have not fixed their intercept to zero. This should be looked at by the authors. Also, in Fig. 5 the x-axis should start from zero.

L283: The authors assert that the 20 m intercepts will give more reliable results than the 100 m intercepts. However, at 100 m the plumes are wider, which offsets somewhat the benefit of greater amplitude of the 20 m intercepts. The authors do not provide an uncertainty analysis currently. The statement here should be justified by demonstrating that the EFs from the 20 m intercepts truly do have lower uncertainties than the 100 m intercepts. The contrast with the background is part of the story, but not the only factor that impacts the uncertainty. For a methods paper, I expect to see more rigorous consideration of measurement uncertainty than is currently provided.

L286: While yes, the observations are "comparable" with other measurements, the authors should certainly note that their measurements are very much on the low end of the literature range.

L290: It is unclear why the authors make their most detailed comparison with Beecken et al., compared to all the other studies cited.

Table 3: The authors need to include Lack et al. (2009, JGR) in their comparison table and in discussion in the text. Lack et al. (2009) report measurements from a variety of different ship types based on plpume intercepts. Their work also clearly shows that the exact EF that one obtains for particles depends on the lower size threshold of the measurement. Here, the authors indicate that it is 10 nm. But, at the same time, the calibration is dependent on the particle size distribution. These issues should certainly be discussed in the context of discussing the measurement accuracy. Perhaps the measured EFs here are on the low side because they really are. But, it may be that some aspect of this is a result of the particular calibration method and the measurement uncertainty. Uncertainties must be discussed more fully, in general.

L299: it is unclear what "in-land transportation" means. Only in looking at the reference is it clear that this means buses operating on "compressed natural gas and ultralow sulfur diesel." It seems that the authors are arguing here that their low PN EF values are a result of the fuel sulfur difference from some of the literature studies. However, I do not find this argument compelling for the simple reason that bus engines are not comparable to marine engines. If the authors want to make this argument, they should compare more directly with ship measurements. For example, the Lack et al. (2011) paper compares PN EF values from before and after a ship in operation switches to low sulfur fuel. They see a negligible difference on the particle number, although the particle mass concentration decreases. This conflicts with the argument that the authors seem to be advancing here through their comparison with a bus study. The same goes for the comparison to the aircraft study. While it is perhaps interesting to compare between engine types, this does not provide any indication that the fuel is what drove this difference.

L311: The authors talk about their method being "validated" because they fall in the range previously observed for ships. To me, this is marginal. A true validation would have used a separate method to measure the EF for this particular ship. This was not done. No discussion of measurement uncertainty has been provided. Thus, we have no way of knowing whether the fact that the measurements here are on the low end of the literature range is because the ship simply had a lower EF or was a result of the measurement itself. For a methods paper, this lacks sufficient details regarding measurement calibration and testing. This is certainly an interesting proof of concept. But, I have substantial concerns regarding the use of terms such as "validation" given the lack of uncertainty analysis or full discussion of specific issues associated with PN measurement using the DISCmini. I think that this paper will only be publishable with a substantially more robust discussion of uncertainties.

Grammar note: The authors consistently say that the "Data was." It should usually be "data are."

---

## Referee Comment (RC2) · Anonymous Referee #3 · 4 Dec 2017

Overall assessment

The research objectives are not clear, is this a proof of concept of the use of UAV systems for emission monitoring and enforcement (MARPOL), or is this climate-related research with the aim to influence maritime policy in that respect, or is this the development of tools to asses emission factors in the field in the optimization of fuel efficiency and emissions in the development of commercial marine engines and emission abating technology (such as scrubbers)?

The authors do not mention some highly relevant projects, studies and operations that have been executed, or are ongoing in Europe whether or not with UAV systems on the

subject of airborne and remote ship emission monitoring. Although the study has some interesting and innovative aspects, the use of UAV systems for emission monitoring is not new and should not be presented as such.

As a proof of concept, a valid comparison should have been made with another (remote) measurement technique, the comparison of the results with emission factors from literature is insufficient as validation.

A significantly more detailed description of the performed calibration methods should have been provided, if the sensors were not adequately calibrated, a data analysis based on the absolute values was not scientifically justified.

A profound and overall accuracy assessment should have been included.

Based on the limited added scientific value, the incomplete presentation of the current related airborne monitoring activities, the insufficient scientific validation and calibration techniques, the lack of an error analysis and some premature assumptions, I recommend to reject the manuscript for publication.

Detailed assessment

Line 23: The authors indicate that emissions were assessed during real world conditions. This is not assessed as such as all measurements were performed from and for one ship. Besides the measured RV is a relatively small vessel (94m) while average merchant vessels are in the order of 200-400m. The RV was also running on ultra low sulphur marine diesel fuel while in reality only a fraction of the international merchant vessels use this fuel type. Different factors may influence the successful assessment of ship emissions among others are: ship-type, ship-age, ship-size, ship-shape, ship-activity, fuel-type, funnel height, funnel shape, wind conditions, inversion layers, etc. For a realistic assessment during real world conditions these factors should have been elaborated. Furthermore for this study the flight path was based on the ship position, in real life ship position is not known in detail, AIS only provides basic navigation info,

e.g. there is no information on the location and shape of the funnel on the ship. The limited autonomy, range and payload of the UAV make this UAV not suitable for realistic operational measurements at sea during real world conditions, the study can therefore hardly be used as a proof of concept. For actual (cost-)effective operations offshore, much more robust fixed- or rotary-wing UAV systems should be used, these systems have other specifications (speed, manoeuvrability etc.) than the one used in this study.

Line 24: The authors indicate that for the first time ship emissions can be assessed and regulated on a reliable and inexpensive way. This is incorrect, as emissions from ships are already assessed and regulated from both airborne, land based and ship-borne sensors in Belgium, The Netherlands, Denmark, Germany and Finland since 2015 at a large scale and on a reliable and cost efficient manner. The use of the UAV's is not necessarily more cost-effective, especially if operated from a ship, and of-ten more time-consuming with less operational output capacity per flight hour. Clearly more information is required to establish cost-effectiveness (platform cost, number of ship measurements per hour, personnel involved, robustness of platform in offshore conditions, ...) . Furthermore the use of UAV's for emission monitoring operations is not new, in 2016 EMSA ordered a feasibility study, granted to CLS, concerning the use of RPAS for emission monitoring (STEAM project), in addition the Danish company EX-PLICIT performed some successful emission measurements with small drones. The only aspect which might be innovative in this study is the measurement of PM emis-sions from ships using drones, but as this is not yet regulated by international law, this has (currently) only academic use.

Line 64: The authors make the assumption that manned aircraft are not feasible for airborne measurements of ship emissions, although the EU funded CompMon project clearly showed the feasibility of manned aircraft for operational emission regulatory airborne surveillance (e.g. operations in Belgium with >2500 monitored ships in 3 years and operations in Denmark with >1000 monitored ships in 2 years).

Line 69: see comment Line 24

Line 142: Sensitivity range for CO2 is 50ppm, this is important as this is same order of magnitude as the delta CO2 for measurements at 100m, this aspect should be discussed further in the article in an overall assessment of the margin of error, which is currently missing.

Line146: Significantly more detailed information should be provided on the calibration method (references samples, calibration-factors, offset, . . .). It is also not clear if a calibration was performed before (and after) every measuring day, this should have been done to ensure the validity of the data.

Line 147 (Figure S1): More information is required for the comparison of the CPC with the DISC, it is not clear what kind of air samples were used for the comparison, it looks like this is just done based on continuous ambient air measurements on board of the RV, for a proper validation a comparison should be made with real emissions. A comparison of the IAQ with the PICARO is completely missing here. If only a comparison (validation) is possible in a lab, this comparison should at least be done during similar conditions as during the field measurement (exposure time, concentration, temperature, . . .), this is clearly not the case as the particle concentrations is very low in this comparison. It looks like the intercept of the linear regression is not put at zero, why is this, was a zero calibration performed? Especially for CO2 it is important to perform the calibration in the same range as the measurement range as the IR absorption is nonlinear, no comments were made on this aspect in the article. Furthermore it should be noted that a linear regression is not an ideal method to compare 2 sensors, the Bland Allman method is more appropriate (Statistical Methods for Assessing Agreement Between Two Methods of Clinical Measurement," by JM Bland and DG Altman, The Lancet, February 8, 1986, 307-310).

Line 158: Flight speed is here expressed as 1.5m/s, it is not clear if this is the airspeed or ground speed. If this is the airspeed, the actual ground speed will depend on the wind conditions, therefore the flight speed through the plume is dependent on the wind conditions too. During the first day, the wind was cross on the ship heading. The plume

would be expected at 180° if transect were flown with alternating heading 250° and 70° (perpendicular to the ship heading), the transect with heading 250° would have been flown with a significant different ground speed (ca. 6.5 m/s instead of 1.5 m/s), no mention is made of this in the article.

Line 208: I would suggest adding an indication of the resulting plume location and flight pattern on the graphs. These graphs would also visualise the different airspeed between the transects (see comment line 158).

Line220: Only 9 times the plume was sampled, very few statistical conclusions can be made based on this small sample size, especially the linear regression on line 277 is questionable.

Line 229: The distance (25m) is missing in this sentence.

Line 232: It is mentioned that the $CO_2$ is up to 100 ppm higher in the plume, this is not clear on the graph (only 50-75 ppm), this will be the part for integration to amount to the delta $CO_2$. Furthermore it should be noted that the peaks for $CO_2$ at a distance of 100 m is of the same order of magnitude of the sensor accuracy.

Line 262: Another flight transect could have been used where the UAV would be flown at the same speed and heading as the RV and hovered in the plume, this would require a transmission of measurement info to the control station to adjust flight altitude and pattern to successfully find the plume and measure the plume for longer periods.

Line280: Instead of a comparison between calculated emission factors and the emission factors from previous studies a comparison with the emission factors calculated based on a plume measurement with the other equipment on board of the RV (e.g. Picaro) would have made more sense.

Line 312: Generalization and misconception that the use of UAV systems would consist of a reduced cost. It is definitely not presented in this article that UAV systems could provide a real cost effective alternative to other surveillance methods as no cost benefit

comparison was made between different surveillance methods (both fixed stations and airborne sensors; operational output capacity; personnel and supporting platform etc.) and all missions were carried out from a vessel, which has a higher operational cost per hour as an aircraft and a lower speed and therefore a much lower cost efficiency.

Note that a higher cost efficiency could maybe be acquired with this setup where these operations would be combined with other task carried out by patrol vessels, pilot ships or research vessels assuming that these vessels would operate within 2 km of shipping lanes. This was not mentioned in the article.

Line 326: $SO_2$ is completely missing here, $SO_2$ is the only emission regulation which is effectively monitored using airborne platforms at this moment and should therefore at least be included in the discussion.

---

## Author Comment (AC1) · 10 Apr 2018

Response to Reviewers Comments: Title: Characterization of the Particle Emission from Ships Operating at Sea Using Unmanned Aerial Vehicles Authors: Tommaso F. Villa, Reece Brown, E. Rohan Jayaratne, L. Felipe Gonzalez, Lidia Morawska, Zoran D. Ristovski The authors thank the Reviewer for the\ comments, and they have modified the manuscript to address them. Reviewer #1: Comment 1: Line 22: There is a typo with the first emission factor given. Answer 1: The typo in the text of the manuscript has been corrected. Line 22 Comment 2: L24: The authors indicate that they have demonstrated a "reliable, inexpensive and accessible" way of measuring ship emissions. The

measurements here required the UAV being deployed from on board the ship. This seems potentially quite limiting. It would be useful if the authors were to rethink the concept of "accessible." I do not dispute that the method here has potential. But, it has not been demonstrated that it is an "accessible" method, especially given the need to optimize the flight path before performing measurements. Answer 2: The authors have taken into consideration the comment of the Reviewers. The use of the UAV system has been defined accessible because it can be deployed from land and specific flight paths can be designed to assess emissions from ships approaching the port area. Such paths can take into consideration different parameters and conditions such as the morphology of the territory, physical barriers and flying speed. This study was a proof of concept and it was decided to deploy form on board the ship to be able to fly away from other ships and without have to obtain permissions from port authorities and civil aviation authorities. In fact this are still the main limiting factors for a large deployment of UAVs. Comment 3: L83: I suggest that both "sophisticated" and "world class" be removed. There is no need for these superlatives, nor are they justified by the description. Answer 3: The authors thank the Reviewer for the comment. The claim has been addressed in the manuscript and the adjectives "sophisticated" and "world class" have been removed from the manuscript. Line 83 Comment 4: Substantailly more information regarding calibration and testing of the DISCmini and the IAQ-calc are needed. The supplemental has no information on the CO2 comparison. This should be added. For the particle comparison, the authors should indicate the measurement conditions. As they note, the calibration depends on the assumed particle size distribution. What was chosen for calibration? Was this just ambient air? Particles produced from an atomizer? Are the calibration particles relevant to the particles in the plume in terms of the size distribution? Were the DISCmini concentrations corrected to account for the difference in slopes in Fig. S1? Is that what is meant by calibration, or are the instruments just being compared? How is uncertainty estimated?

Answer 4: The authors thank the Reviewer for the comment. The following paragraph has been added to the Supplementary material document: "The DISCmini was run in parallel to a CPC 3772 (TSI INCORPORATED 500 Cardigan Road Shoreview, MN, USA) which has a low cut off point of 10 nm. The two instruments were used to sample ambient air from the front mast of the ship. The uncertainty was estimated from the fitting procedure as shown in Figure 5. The IAQ-Calc was placed on the front mast of the ship and the readings were compared to those acquired by the PICARRO spectrophotometer G2301."

Comment 5 :Eqn. 1: The authors use the integrated peak concentrations to calculate the ratio between delta values and the EF. From Fig. 4 it is evident that the $CO_2$ plume is broader than the PN plume. The authors should consider discussing this issue in the context of how it impacts their EF estimates.

Answer 5: The issues with the low amount of data points inside the peak has been addressed in the updated discussion. Page 11; Lines 279-288 "Figure 5 (a) and (b) show the plots of the remaining transects $\Delta$PNC against $\Delta CO_2$ with and without the values of the first flight of day 2. This transect represents a clear outlier in the linear trend, with the $R^2$ value of the linear fit increasing from 0.637 to 0.890 with its exclusion. Furthermore, whilst the linear fit falls within the confidence interval of only one point in (a), it falls within all data points confidence intervals in (b). This occurs despite both $R^2$ values for the fitted Gaussians of this transect being very high ($R^2PNC = 0.9842$, $R^2CO_2 = 0.9518$). This highlights a limitation with this methodology which can be best observed in the difference between Figure 4 (a) and (b). The combination of UAV velocity, sampling rate and response time of the DISCmini results in the PNC transect data having only one data point defining the peak height of the transect. Relying on a single sample point leads to the potential for random instrumentation effects heavily biasing results in a way which does not strongly impact the $R^2$ values of Gaussian fits used to identify successful transects. Therefore, it is unclear whether this is a variation in the ship emissions or an instrumentation error."

Comment 6: L240: Here, the authors focus on differences in absolute values. Such differences can result for a variety of reasons. What really matters, though, is how different the derived EF values are. I suggest that the authors bring the EFs for these plumes into the discussion.

Answer 6: The authors have considered the Reviewer's comment and believe that the claim has been addressed in the discussion paragraph from line 290-297.

Comment 7: Table 2 and Fig. 5: The units given for CO2 are not correct. This must be $kg/m^3$. It is not possible to simply have kg as the units, since the volume is not known. Also, if the units are not $kg/m^3$, then the units on the derived EFs will not make sense.

Answer 7: The authors thank the Reviewer for the comment. The claim has been addressed in the manuscript and both Table 2 and Fig. 5 have been corrected.

Comment 8: Table 2 vs. Fig. 5: There seems to be an inconsistency. The slope from Fig. 5 can be converted to an EF by multiplying by 3.2 kg CO2/kg-fuel. This yields 6.4e15 particles/kg-fuel. But, the average from Table 2 is only 2.6e15 particles/kg-fuel. These should be closer. The difference may be because the authors have not fixed their intercept to zero. This should be looked at by the authors. Also, in Fig. 5 the x-axis should start from zero.

Answer 8: The authors have considered the Reviewer's comment. In the new methodology the final EFPN is calculated using the slope of the line with the intercept fixed at zero as recommended here. The axis has also been updated to start from origin.

Comment 9: L283: The authors assert that the 20 m intercepts will give more reliable results than the 100 m intercepts. However, at 100 m the plumes are wider, which offsets somewhat the benefit of greater amplitude of the 20 m intercepts. The authors do not provide an uncertainty analysis currently. The statement here should be justified by demonstrating that the EFs from the 20 m intercepts truly do have lower uncertainties than the 100 m intercepts. The contrast with the background is part of the story, but not the only factor that impacts the uncertainty. For a methods paper, I expect to see more rigorous consideration of measurement uncertainty than is currently provided.

Answer 9: The Reviewer's comment has been addressed with the methodology being updated. The updated methodology is based on the fitting on Gaussian curves to the transect data in order to find more significant $\Delta$ values. As discussed in the updated manuscript, the broadening of the plume results in significantly poorer Gaussian fits for several reasons. To this end the were excluded from the final calculation of EFPN. This discussion can be found in lines 262-267.

Comment 10: L286: While yes, the observations are "comparable" with other measurements, the authors should certainly note that their measurements are very much on the low end of the literature range.

Answer 10:The updated calculations result in an EFPN more within the range of published values, however can still be considered to be in the lower end. Discussion has been added which address potential reasons for this in lines 290-297 and 308-312.

Comment 11: Table 3: The authors need to include Lack et al. (2009, JGR) in their comparison table and in discussion in the text. Lack et al. (2009) report measurements from a variety of different ship types based on plume intercepts. Their work also clearly shows that the exact EF that one obtains for particles depends on the lower size threshold of the measurement. Here, the authors indicate that it is 10 nm. But, at the same time, the calibration is dependent on the particle size distribution. These issues should certainly be discussed in the context of discussing the measurement accuracy. Perhaps the measured EFs here are on the low side because they really are. But, it may be that some aspect of this is a result of the particular calibration method and the measurement uncertainty. Uncertainties must be discussed more fully, in general.

Answer 11: Lack et al. (2009, JGR) has been added into Table 3. Discussion into the limitations around PNC measurements with the current methodology, included size, has been expanded upon in lines 308-312.

Comment 12: L299: it is unclear what "in-land transportation" means. Only in looking at the reference is it clear that this means buses operating on "compressed natural gas and ultralow sulfur diesel." It seems that the authors are arguing here that their low PN EF values are a result of the fuel sulfur difference from some of the literature studies. However, I do not find this argument compelling for the simple reason that bus engines are not comparable to marine engines. If the authors want to make this argument, they should compare more directly with ship measurements. For example, the Lack et al. (2011) paper compares PN EF values from before and after a ship in operation switches to low sulfur fuel. They see a negligible difference on the particle number, although the particle mass concentration decreases. This conflicts with the argument that the authors seem to be advancing here through their comparison with a bus study. The same goes for the comparison to the aircraft study. While it is perhaps interesting to compare between engine types, this does not provide any indication that the fuel is what drove this difference.

Answer 12:The authors have discussed the Reviewer's comment. In Lack et al. (2011) paper referenced the comparison is between very high sulfur fuel and high sulfur fuel, where reduction in PM mass is observed. The ultra-low sulfur diesel used by the investigator has significantly lower sulfur content than this. In a paper by Ristovski et al. (2006) it was shown the reduction to comparable levels of sulfur content does lead to a reduction in PM number concentration. This has been added into the discussion section. This is elaborated in lines 290-297.

Comment 13: L311: The authors talk about their method being "validated" because they fall in the range previously observed for ships. To me, this is marginal. A true validation would have used a separate method to measure the EF for this particular ship. This was not done. No discussion of measurement uncertainty has been provided. Thus, we have no way of knowing whether the fact that the measurements here are on the low end of the literature range is because the ship simply had a lower EF or was a result of the measurement itself. For a methods paper, this lacks sufficient details regarding measurement calibration and testing. This is certainly an interesting proof of concept. But, I have substantial concerns regarding the use of terms such as "validation" given the lack of uncertainty analysis or full discussion of specific issues associated with PN measurement using the DISCmini. I think that this paper will only be publishable with a substantially more robust discussion of uncertainties.

Answer 13:The updated manuscript attempts to deal with the uncertainties involved in this study with a more robust data analysis and consideration of experimental errors. The word "validated" has been removed as it is agreed that it will be necessary to compare this alongside other developed methodologies before it can be truly validated. Instead, we are treating the study as a proof of concept, and have attempted to highlight the benefits and drawbacks to inform future method development. This has involved many changes across the results and discussion section.

Comment 14: Grammar note: The authors consistently say that the "Data was." It should usually be "data are."

Answer 14: The authors thank the Reviewer for the grammar note, this has been fixed in the updated manuscript. 

Reviewer #3: Comment 1: The authors do not mention some highly relevant projects, studies and operations that have been executed, or are ongoing in Europe whether or not with UAV systems on the subject of airborne and remote ship emission monitoring. Although the study has some interesting and innovative aspects, the use of UAV systems for emission monitoring is not new and should not be resented as such.

Answer 1:The authors considered the Reviewer comment, yet the emphasis was intended to be on the fact that EFPN of ships has never been evaluated with UAVs. The updated manuscript has been modified in multiple lines to clarify this.

Comment 2: Line 23: The authors indicate that emissions were assessed during real world conditions. This is not assessed as such as all measurements were performed from and for one ship. Besides the measured RV is a relatively small vessel (94m) while average merchant vessels are in the order of 200-400m. The RV was also running on ultra low sulphur marine diesel fuel while in reality only a fraction of the international merchant vessels use this fuel type. Different factors may influence the successful assessment of ship emissions among others are: ship-type, ship-age, ship-size, ship-shape, shipactivity, fuel-type, funnel height, funnel shape, wind conditions, inversion layers, etc For a realistic assessment during real world conditions these factors should have been elaborated. Furthermore for this study the flight path was based on the ship position, in real life ship position is not known in detail, AIS only provides basic navigation info e.g. there is no information on the location and shape of the funnel on the ship. The limited autonomy, range and payload of the UAV make this UAV not suitable for realistic operational measurements at sea during real world conditions, the study can therefore hardly be used as a proof of concept. For actual (cost-)effective operations offshore, much more robust fixed- or rotary-wing UAV systems should be used, these systems have other specifications (speed, manoeuvrability etc.) than the one used in this study.

Answer 2: The phrase "real world conditions" is intended to indicate that rather than in a lab or simulated conditions, the UAV was launched on a ship performing operations at sea and measured the exhaust plume. The focus of this paper is on a proof on concept of the methodology. It is not a proof of concept for widespread deployment of this methodology in the field for regulatory or commercial use. That is far beyond the scope of this manuscript. The authors disagree that ship type, class, fuel type, and other differing factors would prevent this methodology from being used. Provided there is an exhaust plume which can be intercepted by the UAV, this methodology can be used to assess emission factors of PNC. The wording of the paper has been changed in multiple places to highlight this is a proof on concept.

Comment 3: Line 24: The authors indicate that for the first time ship emissions can be assessed and regulated on a reliable and inexpensive way. This is incorrect, as emissions from ships are already assessed and regulated from both airborne, land based and shipborne sensors in Belgium, The Netherlands, Denmark, Germany and Finland since 2015 at a large scale and on a reliable and cost efficient manner. The use of the UAV's is not necessarily more cost-effective, especially if operated from a ship, and often more time-consuming with less operational output capacity per flight hour. Clearly more information is required to establish cost-effectiveness (platform cost, number of ship measurements per hour, personnel involved, robustness of platform in offshore conditions, ...) . Furthermore the use of UAV's for emission monitoring operations is not new, in 2016 EMSA ordered a feasibility study, granted to CLS, concerning the use of RPAS for emission monitoring (STEAM project), in addition the Danish company EX-PLICIT performed some successful emission measurements with small drones. The only aspect which might be innovative in this study is the measurement of PM emissions from ships using drones, but as this is not yet regulated by international law, this has (currently) only academic use.

Answer 3: The novel aspect of this paper is the measurement of PM emission factors using a relatively inexpensive UAV. This is primarily is for academic purposes. However, PM has been identified as critical to both health and climate and thus developing the basis for tools which may have potential regulatory of PM emissions is important.

Comment 4: Line 64: The authors make the assumption that manned aircraft are not feasible for airborne measurements of ship emissions, although the EU funded CompMon project clearly showed the feasibility of manned aircraft for operational emission regulatory airborne surveillance (e.g. operations in Belgium with >2500 monitored ships in 3 years and operations in Denmark with >1000 monitored ships in 2 years).

Answer 4: The UAV-based methodology detailed in this manuscript offers an operational setup with orders of magnitude less upfront and operational costs than manned aircraft. The project listed is of a far larger scale and budget than typical research projects.

[Figure]

Comment 5: Line 142: Sensitivity range for CO2 is 50ppm, this is important as this is same order of magnitude as the delta CO2 for measurements at 100m, this aspect should be discussed further in the article in an overall assessment of the margin of error, which is currently missing.

Answer 5:The updated manuscript addresses instrumentation sensitivities and error margins. In particular this comment has been discussed in lines 313-317.

Comment 6: Line146: Significantly more detailed information should be provided on the calibration method (references samples, calibration-factors, offset, : : :). It is also not clear if a calibration was performed before (and after) every measuring day, this should have been done to ensure the validity of the data. Line 147 (Figure S1): More information is required for the comparison of the CPC with the DISC, it is not clear what kind of air samples were used for the comparison, it looks like this is just done based on continuous ambient air measurements on board of the RV, for a proper validation a comparison should be made with real emissions. A comparison of the IAQ with the PICARO is completely missing here. If only a comparison (validation) is possible in a lab, this comparison should at least be done during similar conditions as during the field measurement (exposure time, concentration, temperature,), this is clearly not the case as the particle concentrations is very low in this comparison. It looks like the intercept of the linear regression is not put at zero, why is this, was a zero calibration performed? Especially for CO2 it is important to perform the calibration in the same range as the measurement range as the IR absorption is nonlinear, no comments were made on this aspect in the article. Furthermore it should be noted that a linear regression is not an ideal method to compare 2 sensors, the Bland Allman method is more appropriate (Statistical Methods for Assessing Agreement Between Two Methods of Clinical Measurement," by JM Bland and DG Altman, The Lancet, February 8, 1986, 307-310).

Answer 6: Methodology has been expanded upon significantly in the updated manuscript and a C02 picaro comparison is provided in supplementary material.

Comment 7: Line 158: Flight speed is here expressed as 1.5m/s, it is not clear if this is the airspeed or ground speed. If this is the airspeed, the actual ground speed will depend on the wind conditions, therefore the flight speed through the plume is dependent on the wind conditions too. During the first day, the wind was cross on the ship heading. The plume would be expected at 180_ if ransect were flown with alternating heading 250_ and 70_ (perpendicular to the ship heading), the transect with heading 250_ would have been flown with a significant different ground speed (ca. 6.5 m/s instead of 1.5 m/s), no mention is made of this in the article. Answer 7:Flight speed listed is the airspeed. Whilst the wind conditions will influence the ground speed, the only influence on the measurements will be a variation in the amount of data points captured inside the plume during transect. The discussion of the amount of in plume data points in a transect and its importance is in the updated manuscript in lines 277-281.

Comment 8: Line 208: I would suggest adding an indication of the resulting plume location and flight pattern on the graphs. These graphs would also visualise the different airspeed between the transects (see comment line 158).

Answer 8:The emphasis in the graphs is on the clear detection of the plume by each instrument. The authors do not believe that plume locations would not provide any further information and would overcomplicate the graphs.

Comment 9: Line220: Only 9 times the plume was sampled, very few statistical conclusions can be made based on this small sample size, especially the linear regression on line 277 is questionable.

Answer 9:The methodology has been updated in the updated manuscript.

Comment 10: Line 229: The distance (25m) is missing in this sentence.

Answer 10:This has been clarified in the updated manuscript.

Comment 11: Line 232: It is mentioned that the $CO_2$ is up to 100 ppm higher in the plume, this is not clear on the graph (only 50-75 ppm), this will be the part for integration to amount to the delta CO2. Furthermore it should be noted that the peaks for CO2 at a distance of 100 m is of the same order of magnitude of the sensor accuracy.

Answer 11:The C02 is up to 144ppm counts above background inside the plume in graph 4(a). The graph has been replotted with background removed in the updated manuscript to clarify this. The short 100m transect data has also been discussed in more detail.

Comment 12: Line 262: Another flight transect could have been used where the UAV would be flown at the same speed and heading as the RV and hovered in the plume, this would require a transmission of measurement info to the control station to adjust flight altitude and pattern to successfully find the plume and measure the plume for longer periods.

Answer 12: The focus of this project was the measurement of EFPN through transects of the ship plume. Due to time constraints alternative methodologies could not be investigated, though this suggestion is one of the recommendations for further research listed in the manuscript.

Comment 13: Line280: Instead of a comparison between calculated emission factors and the emission factors from previous studies a comparison with the emission factors calculated based on a plume measurement with the other equipment on board of the RV (e.g. Picaro) would have made more sense.

Answer 13: There was no possibility of accessing the plume with the larger instrumentation such as the picaro or CPC. This is one of the primary advantages of UAV-based platforms. A future validation study would look into this. This is a recommendation in the updated manuscript.

Comment 14: Line 312: Generalization and misconception that the use of UAV systems would consist of a reduced cost. It is definitely not presented in this article that

UAV systems could provide a real cost effective alternative to other surveillance methods as no cost benefit comparison was made between different surveillance methods (both fixed stations and airborne sensors; operational output capacity; personnel and supporting platform etc.) and all missions were carried out from a vessel, which has a higher operational cost per hour as an aircraft and a lower speed and therefore a much lower cost efficiency. Note that a higher cost efficiency could maybe be acquired with this setup where these operations would be combined with other task carried out by patrol vessels, pilot ships or research vessels assuming that these vessels would operate within 2 km of shipping lanes. This was not mentioned in the article.

Answer 14: The authors have addressed this concern in Answer 4, the setup and operational costs of this UAV system are orders of magnitude less than manned aircraft. The focus of this manuscript was on the development of the methodology. Whilst some suggestions for future applications are made, it is premature and beyond the scope of this paper to recommend wide-scale deployments of UAVs and cost benefit comparisons with other methodologies.

Comment 15: Line 326: SO2 is completely missing here, SO2 is the only emission regulation which is effectively monitored using airborne platforms at this moment and should therefore at least be included in the discussion.

Answer 15: The focus of this study was on PN emissions. SO2 would be an interesting alternate application. To the authors knowledge the main challenge for such a system would be that fast and accurate SO2 meters are significantly above the payload of any lightweight UAV, include fixed wings.

Please also note the supplement to this comment:
https://www.atmos-meas-tech-discuss.net/amt-2017-146/amt-2017-146-AC1-supplement.pdf
* * *
**Response to Reviewers Comments:**

Title: Characterization of the Particle Emission from Ships Operating at Sea Using Unmanned

Aerial Vehicles

Authors: Tommaso F. Villa, Reece Brown, E. Rohan Jayaratne, L. Felipe Gonzalez, Lidia

Morawska, Zoran D. Ristovski

The authors thank the Reviewer for the\ comments, and they have modified the manuscript to address them.

**Reviewer #1:**

*Comment 1*: Line 22: There is a typo with the first emission factor given.

*Answer 1:* The typo in the text of the manuscript has been corrected.

Line 22

*Comment 2:* L24: The authors indicate that they have demonstrated a "reliable, inexpensive and
accessible" way of measuring ship emissions. The measurements here required the UAV being
deployed from on board the ship. This seems potentially quite limiting. It would be useful if the
authors were to rethink the concept of "accessible." I do not dispute that the method here has
potential. But, it has not been demonstrated that it is an "accessible" method, especially given the
need to optimize the flight path before performing measurements.

*Answer 2:* The authors have taken into consideration the comment of the Reviewers. The use of
the UAV system has been defined accessible because it can be deployed from land and specific
flight paths can be designed to assess emissions from ships approaching the port area. Such paths
can take into consideration different parameters and conditions such as the morphology of the
territory, physical barriers and flying speed. This study was a proof of concept and it was decided
to deploy form on board the ship to be able to fly away from other ships and without have to

**Fig. 1.** Answer to reviewers comments

[Figure]

**Fig. 2.** Figure 5

Interactive
comment

| Day | Dist/Alt (m) | $R^2_{PNC}$ | $R^2_{CO2}$ | $\Delta PNC$ (#.m$^{-3}$) | $\Delta CO_2$ (kg.m$^{-3}$) | $EF_{PN}$ (#.kg$_{fuel}^{-1}$) |
|-----|--------------|-------------|-------------|---------------------------|------------------------------|--------------------------------|
| 1 | 100/25 | 0.9586 | 0.4998 | 5.05E+11 | 9.35E-05 | 1.73E+16 |
| | 100/35 | 0.4767 | 0.8967 | 4.8E+10 | 1.34E-04 | 1.15E+15 |
| | 20/25 | 0.9856 | 0.8915 | 1.09E+11 | 7.74E-05 | 4.52E+15 |
| 2 | 20/25 | 0.9842 | 0.9518 | 1.06E+12 | 2.83E-04 | 1.20E+16 |
| | 20/25 | 0.9852 | 0.8838 | 3.3E+11 | 1.92E-04 | 5.51E+15 |
| | 20/25 | 0.9489 | 0.9246 | 1.78E+11 | 1.11E-04 | 5.16E+15 |
| | 20/25 | 0.9721 | 0.8965 | 3.6E+11 | 2.23E-04 | 5.18E+15 |
| | 20/25 | 0.9508 | 0.8473 | 1.47E+11 | 1.31E-04 | 3.59E+15 |
| | 20/25 | 0.8517 | 0.6743 | 1.01E+11 | 9.68E-05 | 3.32E+15 |

**Fig. 3.** Table 2

| Reference | Platform | EFPN $(\#.\text{kgfuel}^{-1})$ | Number of ships | Location |
|---|---|---|---|---|
| This Study | UAV | $7.6 \pm 1.4 \times 10^{15}$ | 1 | Open Water |
| Westerlund et al. (2015) | Land Based | $2.35 \pm 0.20 \times 10^{16}$ | 154 | Harbor, Ship Channel |
| Beecken et al. (2014) | Airborne | $1.8 \pm 1.3 \times 10^{16}$ | 174 | Open Water |
| Pirjola et al. (2014) | Land Based | $0.32 \times 10^{16}$ | 11 | Harbor, Ship Channel |
| Alföldy et al (2013) | Land Based | $0.8 \times 10^{16}$ | 497 | Harbor |
| Juwono et al. (2012) | On Board | $0.22 \times 10^{16}$ | 2 | Harbor, Ship Channel |
| Jonsson et al. (2011) | Land Based | $2.55 \pm 0.11 \times 10^{16}$ | 734 | Harbor |
| Lack et al. (2009) | Ship | $0.71 \pm 0.55 \times 10^{16}$ (>13nm)* $1.27 \pm 0.95 \times 10^{16}$ (>5nm)** | 172 165 | Open Water, Shipping Channel |
| Lack et al. (2011) | Airborne | $1.0 \pm 0.2 \times 10^{16}$ | 1 | Open Water |
| Sinha et al. (2003) | Airborne | $6.2 \pm 0.6 \times 10^{16}$ | 2 | Open Water |

**Fig. 4.** Table 3

**Supplement:**

**Supplementary Material (SM) For:**

**2    Characterization of the Particle Emission from Ships Operating**
**3    at Sea Using Unmanned Aerial Vehicles**

Tommaso F. Villa[1], Reece Brown[1], E. Rohan Jayaratne[1], L. Felipe Gonzalez[2], Lidia Morawska[1],
Zoran D. Ristovski[1*]

[1] International Laboratory for Air Quality and Health (ILAQH), Queensland University of Technology (QUT), 2
George St, Brisbane QLD 4000
[2]Australian Research Centre for Aerospace Automation (ARCAA), Queensland University of Technology (QUT), 2
George  St, Brisbane QLD 4000

*Correspondence to*: Zoran D. Ristovski (z.ristovski@qut.edu.au)

The DISCmini was compared with a reference CPC (TSI 3772) for ambient measurements onboard the ship over a period of several hours. Linear regression of the data with an intercept set at origin resulted in an $R^2$ value of 0.982. This regression equation was used to correct DISCmini concentrations in emission factor calculations.

[Figure]

**Figure S1: Comparison of the DISCmini with the CPC in the aerosol laboratory onboard the investigator.**

The IAQ-calc 7545 was compared with a PICARRO Greenhouse Gas Analyzer for ambient measurements onboard the ship over a period of several hours. It was found there was a positive offset of 93 ± 2 ppm (standard error) of between the two measurements.

[Figure]

**Figure S2: Comparison of the IAQ-calc 7545 with the PICARO in the aerosol laboratory onboard the investigator. Bars**
**indicate the 95% confidence interval around the mean.**

---

## Referee Report (RR1)

**General considerations**

**It looks like some major effort has been done to make this article acceptable for publication, but still some major issues remain. I did not have the impression that all changes were included in the last version (version 4), although they have been mentioned to be solved in the author responses. I would request a thorough revision before providing the manuscript again to the reviewers.**

**Analysis of the responses of the author**

*Comment 1 from reviewer 1*: Line 22 : Typo in first emission factor is still not corrected

**Response to the answers from reviewer 3**

*Comment 1*: The authors do not mention some highly relevant projects, studies and operations that have been executed, or are ongoing in Europe whether or not with UAV systems on the subject of airborne and remote ship emission monitoring. Although the study has some interesting and innovative aspects, the use of UAV systems for emission monitoring is not new and should not be resented as such.

*Answer 1:* The authors considered the Reviewer comment, yet the emphasis was intended to be on the fact that EFPN of ships has never been evaluated with UAVs. The updated manuscript has been modified in multiple lines to clarify this.

**Response1: The manuscript is still not clear about the focus on EFPN in several places still "ship emissions" in the broader sense are mentioned without emphasising on EFPN (e.g. in the abstract line 23 and in the conclusion line 336)**

*Comment 2*: Line 23: The authors indicate that emissions were assessed during real world conditions. This is not assessed as such as all measurements were performed from and for one ship. Besides the measured RV is a relatively small vessel (94m) while average merchant vessels are in the order of 200-400m. The RV was also running on ultra low sulphur marine diesel fuel while in reality only a fraction of the international merchant vessels use this fuel type. Different factors may influence the successful assessment of ship emissions among others are: ship-type, ship-age, ship-size, ship-shape, shipactivity, fuel-type, funnel height, funnel shape, wind conditions, inversion layers, etc For a realistic assessment during real world conditions these factors should have been elaborated. Furthermore for this study the flight path was based on the ship position, in real life ship position is not known in detail, AIS only provides basic navigation info e.g. there is no information on the location and shape of the funnel on the ship. The limited autonomy, range and payload of the UAV make this UAV not suitable for realistic operational measurements at sea during real world conditions, the study can therefore hardly be used as a proof of concept. For actual (cost-) effective operations offshore, much more robust fixed- or rotary-wing UAV systems should be used, these systems have other specifications (speed, manoeuvrability etc.) than the one used in this study.

*Answer 2:* The phrase "real world conditions" is intended to indicate that rather than in a lab or simulated conditions, the UAV was launched on a ship performing operations at sea and measured the exhaust plume. The focus of this paper is on a proof on concept of the methodology. It is not a proof of concept for widespread deployment of this methodology in the field for regulatory or

commercial use. That is far beyond the scope of this manuscript. The authors disagree that ship type, class, fuel type, and other differing factors would prevent this methodology from being used. Provided there is an exhaust plume which can be intercepted by the UAV, this methodology can be used to assess emission factors of PNC. The wording of the paper has been changed in multiple places to highlight this is a proof on concept.

**Response 2: An explanations on "the real world conditions" and "proof of concept" should be included in the text, as the text is not clear on this phrasing. The reviewer agrees that as a proof of concept this method is promising, but stating that this method "provides a reliable inexpensive and accessible way to asses and potentially regulate ship emissions" as mentioned in line 23 is a premature conclusion (see Comment 3)**

*Comment 3*: Line 24: The authors indicate that for the first time ship emissions can be assessed and regulated on a reliable and inexpensive way. This is incorrect, as emissions from ships are already assessed and regulated from both airborne, land based and shipborne sensors in Belgium, The Netherlands, Denmark, Germany and Finland since 2015 at a large scale and on a reliable and cost efficient manner. The use of the UAV's is not necessarily more cost-effective, especially if operated from a ship, and often more time-consuming with less operational output capacity per flight hour. Clearly more information is required to establish cost-effectiveness (platform cost, number of ship measurements per hour, personnel involved, robustness of platform in offshore conditions, …) . Furthermore the use of UAV's for emission monitoring operations is not new, in 2016 EMSA ordered a feasibility study, granted to CLS, concerning the use of RPAS for emission monitoring (STEAM project), in addition the Danish company EXPLICIT performed some successful emission measurements with small drones. The only aspect which might be innovative in this study is the measurement of PM emissions from ships using drones, but as this is not yet regulated by international law, this has (currently) only academic use.

*Answer 3:* The novel aspect of this paper is the measurement of PM emission factors using a relatively inexpensive UAV. This is primarily is for academic purposes. However, PM has been identified as critical to both health and climate and thus developing the basis for tools which may have potential regulatory of PM emissions is important.

**Response 3: The abstract in version 4 of the manuscript was not changed on this matter, the reviewer request rephrasing the abstract with emphasis on the novel aspects of PM monitoring using UAV.**

**-UAV's have and are being used for ship emission measurements, so no claim can be made that this is done for the first time.**

**-Accessibility can hardly be claimed as all operations were performed from an RV to measure the RV itself (it would have been possible if test were conducted from shore to measure ships inbound of ports)**

**-Reliability has to be assessed on a much wider sample size, this was not assed, neither mentioned in this proof of concept, so no conclusion on reliability can be drawn on reliability.**

**-No cost efficiency analysis has been made to use this proof of concept for ship emission monitoring (others than the ship it is operating from), therefore it is premature to call this method inexpensive**

**-The author mentions "potentially regulate", as no regulation is currently in place for PM from ships, it is premature to mention this**

*Comment 4*: Line 64: The authors make the assumption that manned aircraft are not feasible for airborne measurements of ship emissions, although the EU funded CompMon project clearly showed the feasibility of manned aircraft for operational emission regulatory airborne surveillance (e.g. operations in Belgium with >2500 monitored ships in 3 years and operations in Denmark with >1000 monitored ships in 2 years).

*Answer 4:* The UAV-based methodology detailed in this manuscript offers an operational setup with orders of magnitude less upfront and operational costs than manned aircraft. The project listed is of a far larger scale and budget than typical research projects.

**Response 4: Line 69 still mentions the limited feasibility of manned aircraft for ship emission monitoring. Note that in Belgium more than 1000 ships per year are monitored for FSC (via measurement of SO2 and CO2). So claiming that manned aircraft are not feasible is not correct. Deploying drones from vessels to inspect the same amount of ships in the same area (in shipping lanes at ca. 60 km from shore) would have been possible, but at significant higher cost. The Belgian aircraft is able to provide a measurement for less than 200€ per inspection, navigating a ship to deploy a drone will cost a multitude of this amount. Drones have a commentary value in for instance the monitoring of inland waters, ports, or when shipping lanes cross near shore, this nuance was not made, simply stating that aircraft are expensive compared to drones is not correct as this requires a description of the operational framework and the application. Furthermore, large scale drones with sufficient radius require similar budgets than manned aircraft and face similar limitations in matter of flying restrictions.**

**The authors mention risks as well as a reason why manned aircraft have limited feasibility, like for all flying operations risks are involved and have to be assessed in a risk analysis, therefore the risks have to be considered indeed, but on the other hand do not necessarily limit the feasibility.**

*Comment 5*: Line 142: Sensitivity range for CO2 is 50ppm, this is important as this is same order of magnitude as the delta CO2 for measurements at 100m, this aspect should be discussed further in the article in an overall assessment of the margin of error, which is currently missing.

*Answer 5:* The updated manuscript addresses instrumentation sensitivities and error margins. In particular this comment has been discussed in lines 313-317.

**Response: The discussion in the text from line 324 to 328 now sufficiently describes this issue**

*Comment 6*: Line146: Significantly more detailed information should be provided on the calibration method (references samples, calibration-factors, offset, : : :). It is also not clear if a calibration was performed before (and after) every measuring day, this should have been done to ensure the validity of the data. Line 147 (Figure S1): More information is required for the comparison of the CPC with the DISC, it is not clear what kind of air samples were used for the comparison, it looks like this is just

done based on continuous ambient air measurements on board of the RV, for a proper validation a comparison should be made with real emissions. A comparison of the IAQ with the PICARO is completely missing here. If only a comparison (validation) is possible in a lab, this comparison should at least be done during similar conditions as during the field measurement (exposure time, concentration, temperature,), this is clearly not the case as the particle concentrations is very low in this comparison. It looks like the intercept of the linear regression is not put at zero, why is this, was a zero calibration performed? Especially for CO2 it is important to perform the calibration in the same range as the measurement range as the IR absorption is nonlinear, no comments were made on this aspect in the article. Furthermore it should be noted that a linear regression is not an ideal method to compare 2 sensors, the Bland Allman method is more appropriate (Statistical Methods for Assessing Agreement Between Two

Methods of Clinical Measurement," by JM Bland and DG Altman, The Lancet, February 8, 1986, 307-310).

*Answer 6:* Methodology has been expanded upon significantly in the updated manuscript and a C02 picaro comparison is provided in supplementary material.

**Response6: Figure S2 does not use a linear regression like for the comparison between the PM sensors. An explanation is missing why a different comparison method was used.**

**Line 151 in the CO2 section refers to Figure S1 instead of figure S2.**

**Figure S2 shows the extensive variation and measurement difference between the Picaro (393ppm) and the IAQ-Calc (486.5ppm), this issue should have been elaborated in the text (see Comment 6). Please explain why the authors did not choose to conduct a calibration with a set of reference span gasses (including zero gas) as this is the most common way to calibration gas analysing sensors?**

**The sensor comparison in Figure S1 uses a linear regression (that intercepts at 0) therefore a calibration can be done of the DISC data using the regression coefficient as the main calibration factor, for the IAQ only an offset is used (93) and no calibration factor, please explain why this different approach was used.**

*Comment 7*: Line 158: Flight speed is here expressed as 1.5m/s, it is not clear if this is the airspeed or ground speed. If this is the airspeed, the actual ground speed will depend on the wind conditions, therefore the flight speed through the plume is dependent on the wind conditions too. During the first day, the wind was cross on the ship heading. The plume would be expected at 180° if transect were flown with alternating heading 250° and 70° (perpendicular to the ship heading), the transect with heading 250° would have been flown with a significant different ground speed (ca. 6.5 m/s instead of 1.5 m/s), no mention is made of this in the article.

*Answer 7:*Flight speed listed is the airspeed. Whilst the wind conditions will influence the ground speed, the only influence on the measurements will be a variation in the amount of data points captured inside the plume during transect. The discussion of the amount of in plume data points in a transect and its importance is in the updated manuscript in lines 277-281.

**Response7: Wind conditions will have an impact in the usability of this method as they impact the number of data points but also the concentrations in the plume (dilution) and therefore depending on the wind condition the measurement could fall outside sensor sensitivity. Furthermore the transect and flight path have to be adjusted depending on the wind conditions, eg. in case of cross wind the measurements behind the ship would not make any sense.**

*Comment 8*: Line 208: I would suggest adding an indication of the resulting plume location and flight pattern on the graphs. These graphs would also visualise the different airspeed between the transects (see comment line 158).

*Answer 8:*The emphasis in the graphs is on the clear detection of the plume by each instrument. The authors do not believe that plume locations would not provide any further information and would overcomplicate the graphs.

**Response 8: Plotting the estimated location of the smoke plume could visualise the issue that was explained in response 7.**

*Comment 9*: Line220: Only 9 times the plume was sampled, very few statistical conclusions can be made based on this small sample size, especially the linear regression on line 277 is questionable.

*Answer 9:*The methodology has been updated in the updated manuscript.

**Response 9: The reviewer argues that the limited number of successful measurements was not sufficiently considered in the manuscript (see comment 3).**

*Comment 10*: Line 229: The distance (25m) is missing in this sentence.

*Answer 10:*This has been clarified in the updated manuscript.

**Response 10: This is sufficiently clarified by the authors**

*Comment 11*: Line 232: It is mentioned that the $CO_2$ is up to 100 ppm higher in the plume, this is not clear on the graph (only 50-75 ppm), this will be the part for integration to amount to the delta $CO_2$. Furthermore it should be noted that the peaks for $CO_2$ at a distance of 100 m is of the same order of magnitude of the sensor accuracy.

*Answer 11:*The $CO_2$ is up to 144ppm counts above background inside the plume in graph 4(a). The graph has been replotted with background removed in the updated manuscript to clarify this. The short 100m transect data has also been discussed in more detail.

**Response 11: This is sufficiently clarified by the authors**

*Comment 12*: Line 262: Another flight transect could have been used where the UAV would be flown at the same speed and heading as the RV and hovered in the plume, this would require a transmission of measurement info to the control station to adjust flight altitude and pattern to successfully find the plume and measure the plume for longer periods.

*Answer 12:* The focus of this project was the measurement of EFPN through transects of the ship plume. Due to time constraints alternative methodologies could not be investigated, though this suggestion is one of the recommendations for further research listed in the manuscript.

**Response 12: This is sufficiently clarified by the authors**

*Comment 13*: Line280: Instead of a comparison between calculated emission factors and the emission factors from previous studies a comparison with the emission factors calculated based on a plume measurement with the other equipment on board of the RV (e.g. Picaro) would have made more sense.

*Answer 13:* There was no possibility of accessing the plume with the larger instrumentation such as the picaro or CPC. This is one of the primary advantages of UAV-based platforms. A future validation study would look into this. This is a recommendation in the updated manuscript.

**Response 13: This is sufficiently clarified by the authors**

*Comment 14*: Line 312: Generalization and misconception that the use of UAV systems would consist of a reduced cost. It is definitely not presented in this article that UAV systems could provide a real cost effective alternative to other surveillance methods as no cost benefit comparison was made between different surveillance methods (both fixed stations and airborne sensors; operational output capacity; personnel and supporting platform etc.) and all missions were carried out from a vessel, which has a higher operational cost per hour as an aircraft and a lower speed and therefore a much lower cost efficiency. Note that a higher cost efficiency could maybe be acquired with this setup where these operations would be combined with other task carried out by patrol vessels, pilot ships or research vessels assuming that these vessels would operate within 2 km of shipping lanes. This was not mentioned in the article.

*Answer 14:* The authors have addressed this concern in Answer 4, the setup and operational costs of this UAV system are orders of magnitude less than manned aircraft. The focus of this manuscript was on the development of the methodology. Whilst some suggestions for future applications are made, it is premature and beyond the scope of this paper to recommend wide-scale deployments of UAVs and cost benefit comparisons with other methodologies.

**Response 14: The reviewer does not want to argue that this cost efficiency comparison should have been made, but claiming that UAV's are less expensive and more effective than other methods also requires a description off the operational framework. The reviewer agrees that for the concept of this research a UAV was most likely less expensive than an aircraft as the RV was monitored during another campaign. Concerning the proximity to the plume, note that the Danish company Explicit is conducting helicopter measurements and is also involved in the development of sensors for drones, in their operational procedures drones fly further away from the ships than the helicopter. The text from line 335 to 339 should be changed as this is based on unfounded assumptions that do not reflect the reality of airborne measurements.**

*Comment 15*: Line 326: SO2 is completely missing here, SO2 is the only emission regulation which is effectively monitored using airborne platforms at this moment and should therefore at least be included in the discussion.

*Answer 15:* The focus of this study was on PN emissions. SO2 would be an interesting alternate application. To the authors knowledge the main challenge for such a system would be that fast  and accurate SO2 meters are significantly above the payload of any lightweight UAV, include  fixed wings.

**Response 15: The sensor-system from Explicit is able to measure SO2, CO2 and NO. This sensor system is in the order of magnitude of a few kg and is deployed on drones and helicopters (slow time response requires monitoring in smoke plumes for up to 30 sec). The sensor used by the Belgian coastguard measures SO2 and CO2, weights 40 kg and has conducted more than 3000 measurements, this sensor uses a Thermo 43I TLE that has been specially modified for faster time response (1-2 sec)**

---

## Editor Decision (ED1)

[revised manuscript text omitted]

---

## Author Response (AR2)

**General considerations**

It looks like some major effort has been done to make this article acceptable for publication, but still
some major issues remain. I did not have the impression that all changes were included in the last
version (version 4), although they have been mentioned to be solved in the author responses. I
would request a thorough revision before providing the manuscript again to the reviewers.

**Analysis of the responses of the author**

*Comment 1 from reviewer 1*: Line 22 : Typo in first emission factor is still not corrected

*Answer 1*: **Emission factor value has been corrected in Line 22**

**Response to the answers from reviewer 3**

*Comment 1*: The authors do not mention some highly relevant projects, studies and operations that
have been executed, or are ongoing in Europe whether or not with UAV systems on the subject of
airborne and remote ship emission monitoring. Although the study has some interesting and
innovative aspects, the use of UAV systems for emission monitoring is not new and should not be
resented as such.

*Answer 1:* The authors considered the Reviewer comment, yet the emphasis was intended to be on
the fact that EFPN of ships has never been evaluated with UAVs. The updated manuscript has been
modified in multiple lines to clarify this.

**Response1:** The manuscript is still not clear about the focus on EFPN in several places still "ship
emissions" in the broader sense are mentioned without emphasising on EFPN (e.g. in the abstract
line 23 and in the conclusion line 336)

**Response Answer 1: Line 24 and line 333 have been clarified.**

*Comment 2*: Line 23: The authors indicate that emissions were assessed during real world
conditions. This is not assessed as such as all measurements were performed from and for one ship.
Besides the measured RV is a relatively small vessel (94m) while average merchant vessels are in the
order of 200-400m. The RV was also running on ultra low sulphur marine diesel fuel while in reality
only a fraction of the international merchant vessels use this fuel type. Different factors may
influence the successful assessment of ship emissions among others are: ship-type,  ship-age,
shipsize, ship-shape, shipactivity, fuel-type, funnel height, funnel shape, wind  conditions, inversion
layers, etc For a realistic assessment during real world conditions these  factors should have been elaborated. Furthermore for this study the flight path was based on the ship position, in real life ship position is not known in detail, AIS only provides basic navigation info e.g. there is no information on the location and shape of the funnel on the ship. The limited autonomy, range and payload of the

UAV make this UAV not suitable for realistic operational measurements at sea during real world conditions, the study can therefore hardly be used as a proof of concept. For actual (cost) effective operations offshore, much more robust fixed- or  rotary-wing UAV systems should be used, these systems have other specifications (speed,  manoeuvrability etc.) than the one used in this study.

*Answer 2:* The phrase "real world conditions" is intended to indicate that rather than in a lab or simulated conditions, the UAV was launched on a ship performing operations at sea and measured the exhaust plume. The focus of this paper is on a proof on concept of the methodology. It is not a proof of concept for widespread deployment of this methodology in the field for regulatory or commercial use. That is far beyond the scope of this manuscript. The authors disagree that ship type, class, fuel type, and other differing factors would prevent this methodology from being used.

Provided there is an exhaust plume which can be intercepted by the UAV, this methodology can be used to assess emission factors of PNC. The wording of the paper has been changed in multiple places to highlight this as a proof of concept.

**Response 2:** An explanations on "the real world conditions" and "proof of concept" should be included in the text, as the text is not clear on this phrasing. The reviewer agrees that as a proof of concept this method is promising, but stating that this method "provides a reliable inexpensive and accessible way to asses and potentially regulate ship emissions" as mentioned in line 23 is a premature conclusion (see Comment 3)

**Response Answer 2:** The authors believe the meaning of "real world conditions" and "proof of concept" are self-evident given the context provided in the paper. The word 'reliable' has been removed from line 24. The authors maintain that the methodology described is inexpensive and accessible in comparison to other methodologies currently available.

*Comment 3*: Line 24: The authors indicate that for the first time ship emissions can be assessed and regulated on a reliable and inexpensive way. This is incorrect, as emissions from ships are already assessed and regulated from both airborne, land based and shipborne sensors in Belgium, The

Netherlands, Denmark, Germany and Finland since 2015 at a large scale and on a reliable and cost efficient manner. The use of the UAV's is not necessarily more cost-effective, especially if operated from a ship, and often more time-consuming with less operational output capacity per flight hour.

Clearly more information is required to establish cost-effectiveness (platform cost, number of ship measurements per hour, personnel involved, robustness of platform in offshore conditions, …) .

Furthermore the use of UAV's for emission monitoring  operations is not new, in 2016 EMSA ordered a feasibility study, granted to CLS, concerning the  use of RPAS for emission monitoring (STEAM

project), in addition the Danish company  EXPLICIT performed some successful emission measurements with small drones. The only aspect which might be innovative in this study is the measurement of PM emissions from ships using drones, but as this is not yet regulated by international law, this has (currently) only academic use.

*Answer 3:* The novel aspect of this paper is the measurement of particle number (PN) emission factors using a relatively inexpensive UAV. This is primarily for academic purposes. However, PN has been identified as critical to both health and climate and thus developing the basis for tools which may suggest that potential regulatory of PN emissions is important.

**Response 3**: The abstract in version 4 of the manuscript was not changed on this matter, the reviewer request rephrasing the abstract with emphasis on the novel aspects of PM monitoring using UAV.

-UAV's have and are being used for ship emission measurements, so no claim can be made that this is done for the first time.

-Accessibility can hardly be claimed as all operations were performed from an RV to measure the RV

itself (it would have been possible if test were conducted from shore to measure ships inbound of ports)

-Reliability has to be assessed on a much wider sample size, this was not assed, neither mentioned in this proof of concept, so no conclusion on reliability can be drawn on reliability.

-No cost efficiency analysis has been made to use this proof of concept for ship emission monitoring (others than the ship it is operating from), therefore it is premature to call this method inexpensive

-The author mentions "potentially regulate", as no regulation is currently in place for PM from ships, it is premature to mention this

**Response Answer 3:**

**-** This is the first time a UAV has been used for $EF_{PN}$ measurements, this has been clarified as discussed previously.

-   The point of departure and landing of the UAV bears no impact on the methodology described
provided that the ship is in flight range. A land base or small boat could be easily used, hence
accessibility. The authors feel that this has been made clear in the paper.
-   The authors agree, reliability has been removed from line 24
-   The upfront cost of this setup is orders of magnitude less than a fixed wing aircraft. Operation of
the UAV requires no fuel and a pilot with relatively minimal training in comparison to a fixed
wing aircraft. The authors feel that this warrants the use of the term "inexpensive" and that a
cost efficiency analysis is irrelevant to the focus of this manuscript.
-   The authors agree and have removed the points on regulation from the manuscript.

*Comment 4*: Line 64: The authors make the assumption that manned aircraft are not feasible for
airborne measurements of ship emissions, although the EU funded CompMon project clearly showed
the feasibility of manned aircraft for operational emission regulatory airborne surveillance (e.g.
operations in Belgium with >2500 monitored ships in 3 years and operations in Denmark with >1000
monitored ships in 2 years).

*Answer 4:* The UAV-based methodology detailed in this manuscript offers an operational setup with
orders of magnitude less upfront and operational costs than manned aircraft. The project listed is of
a far larger scale and budget than typical research projects.

**Response 4**: Line 69 still mentions the limited feasibility of manned aircraft for ship emission
monitoring. Note that in Belgium more than 1000 ships per year are monitored for FSC (via
measurement of SO2 and CO2). So claiming that manned aircraft are not feasible is not correct.
Deploying drones from vessels to inspect the same amount of ships in the same area (in shipping
lanes at ca. 60 km from shore) would have been possible, but at significant higher cost. The Belgian
aircraft is able to provide a measurement for less than 200€ per inspection, navigating a ship to
deploy a drone will cost a multitude of this amount. Drones have a commentary value in for instance
the monitoring of inland waters, ports, or when shipping lanes cross near shore, this nuance was not
made, simply stating that aircraft are expensive compared to drones is not correct as this requires a
description of the operational framework and the application. Furthermore, large scale drones with
sufficient radius require similar budgets than manned aircraft and face similar limitations in matter
of flying restrictions.

The authors mention risks as well as a reason why manned aircraft have limited feasibility, like for all
flying operations risks are involved and have to be assessed in a risk analysis, therefore the risks have
to be considered indeed, but on the other hand do not necessarily limit the feasibility.

**Response Answer 4:**

The aircraft discussed measures SO2 and CO2 which are gaseous pollutants not relevant to the EFPNs which are the focus of this methodology. It is likely this aircraft can only be operated viably at such a cost due to the high demand of measuring these pollutants for regulatory applications. Measurement of PN emissions for research applications are in much lower demand, hence why there are no dedicated manned aircraft available for these measurements at such a cost. This aside, the reviewer has indicated several scenarios in which manned vehicles are not viable, supporting the authors claim that they have limited feasibility.

*Comment 5*: Line 142: Sensitivity range for CO2 is 50ppm, this is important as this is same order of magnitude as the delta CO2 for measurements at 100m, this aspect should be discussed further in the article in an overall assessment of the margin of error, which is currently missing.

*Answer 5:* The updated manuscript addresses instrumentation sensitivities and error margins. In particular this comment has been discussed in lines 313-317.

**Response 5:** The discussion in the text from line 324 to 328 now sufficiently describes this issue

*Comment 6*: Line146: Significantly more detailed information should be provided on the calibration method (references samples, calibration-factors, offset, : : :). It is also not clear if a calibration was performed before (and after) every measuring day, this should have been done to ensure the validity of the data. Line 147 (Figure S1): More information is required for the comparison of the CPC with the DISC, it is not clear what kind of air samples were used for the comparison, it looks like this is just done based on continuous ambient air measurements on board of the RV, for a proper validation a comparison should be made with real emissions. A comparison of the IAQ with the PICARO is completely missing here. If only a comparison (validation) is possible in a lab, this comparison should at least be done during similar conditions as during the field measurement (exposure time, concentration, temperature,), this is clearly not the case as the particle concentrations is very low in this comparison. It looks like the intercept of the linear regression is not put at zero, why is this, was a zero calibration performed? Especially for CO2 it is important to perform the calibration in the same range as the measurement range as the IR absorption is nonlinear, no comments were made on this aspect in the article. Furthermore it should be noted that a linear regression is not an ideal method to compare 2 sensors, the Bland Allman method is more appropriate (Statistical Methods for Assessing Agreement Between Two Methods of Clinical Measurement," by JM Bland and DG Altman, The Lancet, February 8, 1986, 307310).

*Answer 6:* Methodology has been expanded upon significantly in the updated manuscript and a C02

picaro comparison is provided in supplementary material.

**Response 6**: Figure S2 does not use a linear regression like for the comparison between the PM

sensors. An explanation is missing why a different comparison method was used.

Line 151 in the CO2 section refers to Figure S1 instead of figure S2.

Figure S2 shows the extensive variation and measurement difference between the Picaro (393ppm) and the IAQ-Calc (486.5ppm), this issue should have been elaborated in the text (see

Comment 6). Please explain why the authors did not choose to conduct a calibration with a set of reference span gasses (including zero gas) as this is the most common way to calibration gas analysing sensors?

The sensor comparison in Figure S1 uses a linear regression (that intercepts at 0) therefore a calibration can be done of the DISC data using the regression coefficient as the main calibration factor, for the IAQ only an offset is used (93) and no calibration factor, please explain why this different approach was used.

**Response Answer 6**

Line 149 has been corrected.

The calibrations of the instrumentation were performed using ambient aerosol measurements. This has been clarified in line 148. The PN concentrations show a linear response trend which allowed for a good fit. As is shown in S2 no such trend was observed for $CO_2$, instead an offset error were calculated through the mean value indicated as shown in the supplementary material.

*Comment 7*: Line 158: Flight speed is here expressed as 1.5m/s, it is not clear if this is the airspeed or ground speed. If this is the airspeed, the actual ground speed will depend on the wind conditions, therefore the flight speed through the plume is dependent on the wind conditions too. During the first day, the wind was cross on the ship heading. The plume would be expected at 180° if transect were flown with alternating heading 250° and 70° (perpendicular to the ship  heading), the transect with heading 250° would have been flown with a significant different  ground speed (ca. 6.5 m/s instead of 1.5 m/s), no mention is made of this in the article.

*Answer 7:* Flight speed listed is the airspeed. Whilst the wind conditions will influence the ground speed, the only influence on the measurements will be a variation in the amount of data points captured inside the plume during transect. The discussion of the amount of in plume data points in a transect and its importance is in the updated manuscript in lines 277-281.

**Response 7**: Wind conditions will have an impact in the usability of this method as they impact the number of data points but also the concentrations in the plume (dilution) and therefore depending on the wind condition the measurement could fall outside sensor sensitivity. Furthermore the transect and flight path have to be adjusted depending on the wind conditions, eg. in case of cross wind the measurements behind the ship would not make any sense.

**Response Answer 7:** The impact of wind direction and ambient condition changes are addressed in

Paragraph 252-260. As stated in Line 158 the RV Investigator was heading into the wind during measurements, therefore the flight path was discussed in terms of position from rear of ship to ensure clear discussion. The authors believe it is self-explanatory that the flight path would need to be changed if the ship was not oriented into the wind.

*Comment 8*: Line 208: I would suggest adding an indication of the resulting plume location and flight pattern on the graphs. These graphs would also visualise the different airspeed between the transects (see comment line 158).

*Answer 8:* The emphasis in the graphs is on the clear detection of the plume by each instrument. The authors do not believe that plume locations would not provide any further information and would overcomplicate the graphs.

**Response 8**: Plotting the estimated location of the smoke plume could visualise the issue that was explained in response 7.

**Response Answer 8:**

The authors believe this issue to be sufficiently discussed and maintain that further information would overcomplicate the graphs.

*Comment 9*: Line220: Only 9 times the plume was sampled, very few statistical conclusions can be made based on this small sample size, especially the linear regression on line 277 is questionable.

*Answer 9:* The methodology has been updated in the updated manuscript.

**Response 9:** The reviewer argues that the limited number of successful measurements was not sufficiently considered in the manuscript (see comment 3).

**Response Answer 9:**

Despite the number of samples being relatively small, there are two aspects that support our assertion that the result is significant and reliable. First, we note that the best line in the graph passes through the error margins of every one of the viable data points collected, suggesting that there is a linear relationship. Secondly, the value of the emission factor calculated using the linear regression is well within the range of values found in a number of other published studies and listed in Table 3. The paper is transparent about the size of the data set used.

*Comment 10*: Line 229: The distance (25m) is missing in this sentence.

*Answer 10:* This has been clarified in the updated manuscript.

**Response 10:** This is sufficiently clarified by the authors

*Comment 11*: Line 232: It is mentioned that the CO2 is up to 100 ppm higher in the plume, this is not clear on the graph (only 50-75 ppm), this will be the part for integration to amount to the delta CO2.

Furthermore it should be noted that the peaks for CO2 at a distance of 100 m is of the same order of magnitude of the sensor accuracy.

*Answer 11:* The C02 is up to 144ppm counts above background inside the plume in graph 4(a). The graph has been replotted with background removed in the updated manuscript to clarify this. The short 100m transect data has also been discussed in more detail.

**Response 11:** This is sufficiently clarified by the authors

*Comment 12*: Line 262: Another flight transect could have been used where the UAV would be flown at the same speed and heading as the RV and hovered in the plume, this would require a transmission of measurement info to the control station to adjust flight altitude and pattern to successfully find the plume and measure the plume for longer periods.

*Answer 12:* The focus of this project was the measurement of EFPN through transects of the ship plume. Due to time constraints alternative methodologies could not be investigated, though this suggestion is one of the recommendations for further research listed in the manuscript.

**Response 12:** This is sufficiently clarified by the authors

*Comment 13*: Line280: Instead of a comparison between calculated emission factors and the emission factors from previous studies a comparison with the emission factors calculated based on a plume measurement with the other equipment on board of the RV (e.g. Picaro) would have made more sense.

*Answer 13:* There was no possibility of accessing the plume with the larger instrumentation such as the picaro or CPC. This is one of the primary advantages of UAV-based platforms. A future validation study would look into this. This is a recommendation in the updated manuscript.

**Response 13:** This is sufficiently clarified by the authors

*Comment 14*: Line 312: Generalization and misconception that the use of UAV systems would consist of a reduced cost. It is definitely not presented in this article that UAV systems could  provide a real cost effective alternative to other surveillance methods as no cost benefit  comparison was made between different surveillance methods (both fixed stations and airborne  sensors; operational output capacity; personnel and supporting platform etc.) and all missions  were carried out from a vessel, which has a higher operational cost per hour as an aircraft and a  lower speed and therefore a much lower cost efficiency. Note that a higher cost efficiency could maybe be acquired with this setup where these operations would be combined with other task carried out by patrol vessels, pilot ships or research vessels assuming that these vessels would operate within 2 km of shipping lanes.

This was not mentioned in the article.

*Answer 14:* The authors have addressed this concern in Answer 4, the setup and operational costs of this UAV system are orders of magnitude less than manned aircraft. The focus of this manuscript was on the development of the methodology. Whilst some suggestions for future applications are made, it is premature and beyond the scope of this paper to recommend wide-scale deployments of UAVs and cost benefit comparisons with other methodologies.

**Response 14**: The reviewer does not want to argue that this cost efficiency comparison should have been made, but claiming that UAV's are less expensive and more effective than other methods also requires a description off the operational framework. The reviewer agrees that for the concept of this research a UAV was most likely less expensive than an aircraft as the RV was monitored during another campaign. Concerning the proximity to the plume, note that the Danish company Explicit is conducting helicopter measurements and is also involved in the development of sensors for drones, in their operational procedures drones fly further away from the ships than the helicopter. The text from line 335 to 339 should be changed as this is based on unfounded assumptions that do not reflect the reality of airborne measurements.

**Response Answer 14:** The authors feel that their claims to the relative inexpensiveness of this system have been explained and are justified. In regards to the comparison between this methodology for the measurement of $EF_{PN}$ for research applications and the commercial measurement of regulated pollutants by private companies, please consider the response given in

Answer 4.

*Comment 15*: Line 326: SO2 is completely missing here, SO2 is the only emission regulation which is effectively monitored using airborne platforms at this moment and should therefore at least be included in the discussion.

*Answer 15:* The focus of this study was on PN emissions. SO2 would be an interesting alternate application. To the authors knowledge the main challenge for such a system would be that fast and accurate SO2 meters are significantly above the payload of any lightweight UAV, include fixed wings.

**Response 15**: The sensor-system from Explicit is able to measure SO2, CO2 and NO. This sensor system is in the order of magnitude of a few kg and is deployed on drones and helicopters (slow time response requires monitoring in smoke plumes for up to 30 sec). The sensor used by the Belgian coastguard measures SO2 and CO2, weights 40 kg and has conducted more than 3000

measurements, this sensor uses a Thermo 43I TLE that has been specially modified for faster time response (1-2 sec)

**Response Answer 15:** While there are a number of pollutants that could have been measured, this study was focussed on particle number concentration, as this is a parameter that has not been monitored in ships plumes using UAVs in the past. Aside from $SO_2$ sensors not being relevant to the calculations of $EF_{PN}$, again we argue that the main challenge for such a system would be that fast and accurate $SO_2$ meters would be significantly above the payload of any lightweight UAV.

[revised manuscript text omitted]

---

## Author Response (AR3)

**Characterization of the Particle Emission from a Ships Operating at Sea Using an 
[revised manuscript text omitted]